# LM-Switch: Transforming Word Embedding Space for Flexible Language Model Steering

## Abstract

Large language models (LLMs) have advanced significantly as general-purpose tools. Varied real-life demands, ranging from risk management for specific audiences to customizing text styles for different scenarios, all necessitate customizing general-purpose LLMs to different *conditions*. However, existing pre-training or fine-tuning solutions are still not efficient or flexible enough, and can compromise LLMs' original quality. Applying classifiers as constraints requires an expensive decoding process. We motivate ourselves by theoretically interpreting the role of word embeddings in modeling output distribution. By analyzing a variant of Hidden Markov Models (HMMs), we find that different conditions in HMMs can be surprisingly understood as linear transformations in the output word embedding space. This finding inspires LM-Switch, a novel, theoretically grounded, lightweight, transferrable, and flexible method for generative language model conditioning. LM-Switch simply deploys a linear transformation in the output word embedding space. It can achieve comparable or superior performance compared with state-of-the-art baselines in LM detoxification and sentiment control while maintaining a better balance with generation quality, despite training only 0.2% of model parameters. It is also able to learn from a few sentences or one document. One can continuously steer LLMs by scaling the transformation, or compose multiple conditions by adding their transformations. Moreover, a learned LM-Switch can be transferred to other LLMs of different sizes. We will make our code available to the research community following publication. [1]

## 1 Introduction

In recent years, large language models (LLMs) have significantly advanced various natural language processing (NLP) tasks such as machine translation, sentiment analysis, schema induction and summarization (Brown et al., 2020; Kojima et al.; Li et al., 2023; Radford et al., 2018; OpenAI, 2023). LLMs are typically pre-trained on a unified text corpus. However, there are various scenarios where it is desirable to steer a language model's generation according to different *conditions*, such as stances, styles, and sentiments. Examples of these cases include tailoring a language model output for specific scenarios (such as academic or social media), managing risks when facing different audiences, or mitigating bias pre-trained into language models, etc. When facing these diverse needs, retraining or fine-tuning is not only inefficient (Brown et al., 2020) but can also negatively impact their performance (Zhang et al., 2022b). This work is motivated by the need for efficiently adapting existing LLMs to diverse *conditions* without extensive retraining or a compromise in performance.

There has been increasing attention on controlling LM generations. Besides directly training an LM on domain-specific datasets (Zhang et al., 2020; Li et al., 2022; Zhang et al., 2018), other techniques are proposed for guiding LM at decoding time. These attempts include superposing attribute classifiers (such as sentiment and toxicity) as constraints when sampling tokens (Kumar et al., 2022; Dathathri et al.; Liu et al., 2021; Yang & Klein, 2021b; Li et al., 2018), treating decoding as an optimization problem (Kumar et al., 2021), or training parameter efficient adaptor modules in existing LMs (Hu et al., 2022; 2021a). Despite these efforts, due to the large size of parameters in LMs to be adapted, and the extra computation burden while decoding, efficient and flexible steering

---

[1]**Please be advised that this paper contains potentially controversial results and examples to some readers, included solely for research purposes to explore model capabilities.**

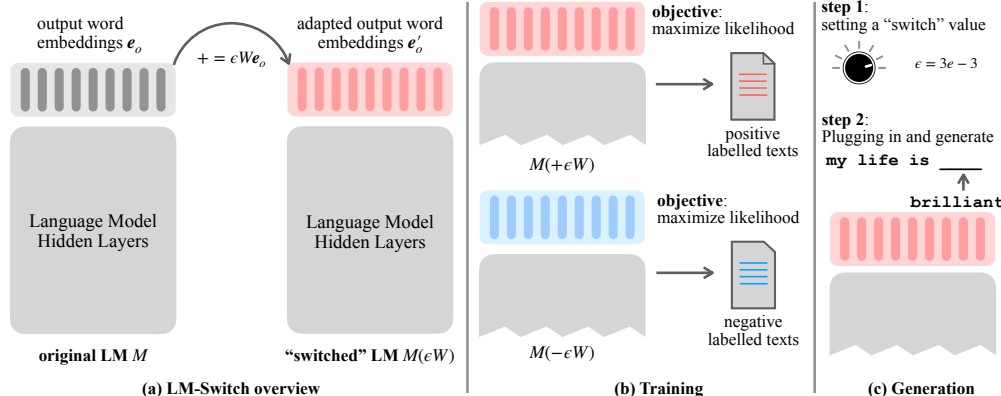

Figure 1: An overview of LM-Switch. **(a)**: LM-Switch applies a linear factor $\epsilon W \mathbf{e}_v$ to each word embedding for language model conditioning. **(b)**: During training, we use a positively switched model $M(\epsilon W)$ to maximize likelihood on positively labelled texts, vise versa. **(c)**: For generation, one only needs to specify a switch value $\epsilon$, and then proceed with normal decoding.

of LMs while not compromising its generaion quality still remains challenging. Recently, prompting with instructions has emerged as a novel method for LM interaction (Brown et al., 2020; OpenAI, 2023). However either the performance relies on emergent instruction following of super large LMs, and/or needs to pre-train an instruction-controlled LM deliberately (Zhou et al., 2023; Raffel et al., 2020), limiting scalability to higher demands and smaller models.

To address these challenges, we introduce LM-Switch, a theoretically grounded yet empirically straightforward and lightweight plug-in for efficient and versatile conditioning over language models, by only transforming the output word embedding space. We start by theoretically investigating the role of word embeddings in text distributions. Typically, we use the dot product between a computed context vector $\mathbf{c}$ and a learnable output word embedding $\mathbf{e}$ as the word logit. Specifically, we find the effect of condition shifts in Hidden Markov Models can be associated with linear transformations on output word embedding in LMs. This inspires the design of our proposed method, LM-Switch, where we apply a $d \times d$ learnable linear transformation $W$ on the output word embeddings, where $d$ is the embedding dimension. Specifically, the embedding $\mathbf{e}_v$ of each word $v \in \mathcal{V}$ is replaced with $\mathbf{e}_v + \epsilon W \mathbf{e}_v$. Here $\epsilon$ acts as a "switching value" to indicate polarity and intensity.

Empirically, LM-Switch achieves comparable or superior performance on language model detoxification and sentiment control, while demonstrating a list of advantages. LM-Switch is both **parameter efficient** (using 1.6M parameters on GPT2-large, which is only 0.2% of LM parameters and 9% of the size of LoRA (Hu et al., 2021a), a parameter-efficient fine-tuning method) and **data-efficient** (able to learn from one article or dozens of sentences). A learned LM-Switch is **transferable** to other LMs with different sizes without additional training, achieving detoxification performance similar to the best baseline. Moreover, LM-Switch enjoys a linearity property theoretically, which enables both **continuous** and **compositional** control. This allows for dealing with diverse and nuanced situations, such as personalized or customized generation, without re-training for each scenario. Moreover, LM-Switch is also **interpretable**, able to reveal the most indicative words and word dimensions associated with a task. In summary, this paper makes the following contributions:

- We propose LM-Switch, a theoretically supported and lightweight method for language model conditioning.
- We empirically demonstrate the effectiveness of LM-Switch on applications such as LM detoxification and sentiment control.
- We also highlight and prove the benefits of LM-Switch including data efficiency, transferability, continuous and compositional control, and interpretability.

## 2 RELATED WORK

**Control of Language Models** has been of growing interest in recent years, motivated by the increasing capabilities of LMs. This area originates from the need to leverage the generation capabilities of

large language models while avoiding the need for time-consuming and costly retraining or fine-tuning. These attempts include applying attribute classifiers or heuristic constraints at decoding time (Kumar et al., 2022; Dathathri et al.; Liu et al., 2021; Yang & Klein, 2021b), treating the generation process as an optimization problem over the embedding or token sequences (Kumar et al., 2021), or post-editing the output (Li et al., 2018). These techniques are often computationally expensive in searching the output and rely on the availability and quality of suitable external classifiers. More recently, prompting-based control for large language models has received much attention, with the control achieved by input prompt engineering to guide the model's generation. However, this method often relies on the quality and availability of large language models (Brown et al., 2020; OpenAI, 2023), and may also necessitate the deliberate training (Raffel et al., 2020; Zhou et al., 2023). It can also be challenging to design effective prompts for complex or nuanced control goals. Parameter-efficient fine-tuning such as LoRA (Hu et al., 2021a) focuses on learning low-rank approximations of model parameters. However, this is still essentially a fine-tuning-based method, and cannot achieve flexible and transferrable language model steering like ours. Probably most closely related to our work are attempts to discover "steering" vectors or tokens (Subramani et al., 2022; Li & Liang, 2021), which might originate from similar work in image generation (Jahanian et al.; Hu et al., 2021b). Different from our model, these efforts focus on other applications such as multi-task learning and sentence recovery, and the learned vectors (instead of matrices as in our work) are not shown to be transferrable or interpretable, nor enable flexible control.

**Controllable Text Generation** is a broader topic involving generating text according to various control objectives such as dialogue history, personality, format or knowledge (Zhang et al., 2022a; Fang et al., 2022; Ke et al., 2022; Yang & Klein, 2021a; Pascual et al., 2021; Liu et al., 2021; Yan et al., 2021; Yu et al., 2022; Zhang et al., 2020). Different from prior work which often requires training a task-specific model, our model mainly focuses on providing plug-and-play conditioning over a diverse range of off-the-shelf language models. Note that some methods among these papers do not support users to provide text suffixes, or "prompting" (Li et al., 2022; Qin et al.; Lu et al., 2021; Post & Vilar, 2018), incompatible with the evaluation setting of this study.

**Language Model Detoxification** Motivated by the goal to address the systematic biases embedded in language models, there are efforts in conducting language model de-biasing or de-toxification (Meade et al., 2022; Kaneko et al., 2022). Approaches span all aspects of the language model pipeline. A line of work focuses on automatically obtaining cleaner data (Barikeri et al., 2021; Webster et al., 2021; Dinan et al., 2020). Another line of work modifies the model workflow design to explicitly accommodate the bias factors (Webster et al., 2021; Schick et al., 2021; Yu et al., 2023; Omrani et al., 2023; Yang et al., 2023). The most related line of work to the herein proposed method involves manipulating embedding space such as Principle Component Analysis and Nullspace Projection (Liang et al., 2020; Bolukbasi et al., 2016; Ravfogel et al., 2020). The evaluation in these settings (Kaneko & Bollegala, 2021; Nadeem et al., 2021; Nangia et al., 2020) mostly consists of quiz-question checking for stereotypical misbeliefs. More related to our method are those mentioned in language model control (Kumar et al., 2022; Dathathri et al.; Liu et al., 2021; Yang & Klein, 2021b; Kumar et al., 2021), which constrains or guides text generation according to a classifier. A unique contribution in our work is that the learned LM-Switch can be transferred to detoxify other off-the-shelf language models without a costly training process.

## 3 LM-Switch: Motivation and Formulation

We first provide a theoretical inspiration of LM-Switch. Hidden Markov Model (HMM) is a widely used framework for analyzing discrete stochastic processes. Because of its generality (modeling arbitrary distributions), intuitiveness, and interpretability (containing a structured state space), it has long been used as a primary choice when modeling language distribution. Our theoretical analysis shows that under certain assumptions switching between conditions is equivalent to a linear transform in word embedding spaces. This observation then inspires the derivation of our proposed model.

### 3.1 Theoretical Motivation of LM-Switch

**Hidden Markov Models** Hidden Markov Models (HMMs) (Baum et al., 1970) is a discrete stochastic process with a set of $n$ states $\mathbf{S}$ and a set of $m$ observations or emissions $\mathbf{O}$, with arbitrary indexing of $\mathbf{S}$ and $\mathbf{O}$. The distribution for the time step $t = 0$ is determined by initial state distribution

$s_0 \sim \pi$. For each later time step $t \geq 1$, the state transition probabilities are represented by a matrix $\mathbf{T}$, where $T(s, s') = P(s_{t+1} = s' | s_t = s)$ denotes the probability of transitioning from state $s$ to state $s'$. At each time step one observation $o_t$ is emitted, with the emission probabilities represented by a matrix $\mathbf{B}$, with $B(s, o) = P(o_t = o | s_t = s)$. A sequence of observations can be denoted as $\mathbf{o} = \{o_1, o_2, \ldots, o_T\}$. The probability distribution over sequences $\mathbf{o}$ then follows formula:

$$P(o_1, \cdots, o_T; \pi) = \pi^\top \left( \prod_{t=0}^{T-1} \mathrm{diag}(\mathbf{p}(o_t)) T \right) \mathbf{p}(o_T), \tag{1}$$

where $\mathbf{p}(o)$ is a $|\mathcal{S}|$-dim vector indicating $P(o \mid s)$ for all states $s \in \mathcal{S}$.

**Language Models** In generative language models, the sequence is generated word-by-word by a conditional probability $P(o_t \mid o_1, \cdots, o_{t-1})$. The common technique to model this probability is to first calculate the inner product between a contextual vector $\mathbf{c}(o_1, \cdots, o_{t-1})$ and word embeddings $\mathbf{E} = (\mathbf{e}_o, \cdots) \in \mathbb{R}^{d \times |\mathcal{O}|}$, namely, $\mathbf{l} = \mathbf{c}(o_1, \cdots, o_{t-1})^\top \mathbf{E}$. Here, $\mathbf{l}$ is known as the word *logits*, which then usually passes through a softmax operator to get a distribution over words. For simplicity of analysis, in this work, we assume a linear formulation and let conditional probability $P(o_t | o_1, \cdots, o_{t-1}) = \mathbf{c}(o_1, \cdots, o_{t-1})^\top \mathbf{e}_{o_t}$. By the chain rule, multiplying the conditional probabilities will give us the full probability: $\prod_{t=1}^{T} P(o_t \mid o_1, \cdots, o_{t-1}) = P(o_1, \cdots, o_T)$. We are then interested in the situation where a language model is good enough to represent an equivalent distribution with HMM.

**Relating Transforming Word Embeddings with LM Steering** In this study, we aim to model the influence of linearly transforming word embeddings in text generation. As a theoretical motivation, we first show that, certain shifts in an HMM's state initialization can be equivalent to a linear transformation in word embedding space. We leave the more formal statement as well as the proof of this result to Appendix B, and present an intuitive interpretation here.

**Theorem 1.** *Under certain constraints, in a low-rank HMM, shifting from one state initialization to another is equivalent to a linear transformation in word embedding space.*

### 3.2 LM-SWITCH FORMULATION

Inspired by the discovery in the section above, we propose LM-Switch to apply a linear transform in the output word embedding space. LM-Switch is conceptually simple and straightforward to implement. An illustration of LM-Switch is presented in Figure 1(a). Specifically, let $M$ be a fixed language model with fixed parameters. We replace its each output word embeddings $\mathbf{e}_v$ with $\mathbf{e}_v + \epsilon W \mathbf{e}_v$, and call the resulting language model $M' = M(\epsilon W)$ a "switched model". Here the "switch matrix" $W$ is the only learnable parameter determining the effect of LM-Switch, and $\epsilon$ is a manually adjustable scalar indicating the polarity and intensity of the "switch value". Without loss of generality, we arbitrarily pick a small value $\epsilon_0 = 1e - 3$ as the default switch value.[2] We use $P(\mathbf{o}|\epsilon W)$ to denote the probability of $M'$ generating sequence $\mathbf{o} = (o_1, \cdots, o_T)$. Figure 1(b, c) shows the training and generation process of LM-Switch. During training, we use the positively switched model $M(\epsilon W)$ to fit the positively labeled texts, with maximal likelihood as the training objective. When negative texts are available, we also fit them with $M(-\epsilon W)$. When generating with LM-Switch, the user only needs to specify a switch value $\epsilon$ and then decode the language model. More details are in Appendix D. Intuitively explaining, LM-Switch learns to identify word embedding dimensions that are best associated with a target condition, and manipulate among those dimensions to achieve language rewording. In Section 5, we show this enables an interpretation of word embeddings by analyzing the learned $W$.

We also theoretically and empirically compare LM-Switch against a simplified version, a **(soft) word blacklist (SWB)**: a (learnable) logit offset is applied to each token candidate after the original logits are computed. As we demonstrate in Appendix K, adding a (learnable) vector to context vectors $\mathbf{c}$ achieves a similar effect with SWB. We also further prove that LM-Switch is theoretically expressive of any distribution shift, while a SWB is unable to do so. In 4 we show that SWB indeed yields inferior performance than LM-Switch.

---

[2]Using $\epsilon W$ achieves the equivalent effect as $k\epsilon \cdot k^{-1}W)$ when $k \neq 0$, so the absolute value of $\epsilon$ itself is only meaningful when also considering the magnitude of $W$.

Table 1: On language model detoxification, LM-Switch achieves best performance.

| Model | Backbone Size | Toxicity↓ | | Fluency | Diversity↑ | | |
|---|---|---|---|---|---|---|---|
| | | Max. toxicity | Toxicity prob. | Output ppl.↓ | Dist-1 | Dist-2 | Dist-3 |
| PPLM (10%) | 345M | 0.520 | 0.518 | 32.58 | 0.58 | 0.86 | 0.86 |
| DAPT | 110M | 0.428 | 0.360 | 31.21 | 0.57 | 0.84 | 0.84 |
| GeDi | 1.5B | 0.363 | 0.217 | 60.03 | 0.62 | 0.84 | 0.83 |
| DExperts$_{base}$ | 117M | 0.302 | 0.118 | 38.20 | 0.56 | 0.82 | 0.83 |
| DExperts$_{medium}$ | 340M | 0.307 | 0.125 | 32.51 | 0.57 | 0.84 | 0.84 |
| DExperts$_{large}$ | 762M | 0.314 | 0.128 | 32.41 | 0.58 | 0.84 | 0.84 |
| PromptT5 | 780M | 0.320 | 0.172 | 354.71 | 0.58 | 0.76 | 0.70 |
| MuCoLa | 762M | 0.308 | 0.088 | 29.92 | 0.55 | 0.82 | 0.83 |
| LoRA | 762M | 0.365 | 0.210 | 21.11 | 0.53 | 0.85 | 0.86 |
| Soft-Blacklist | 762M | 0.270 | 0.154 | 18.28 | 0.53 | 0.81 | 0.83 |
| LM-Switch$_{base}$ | 117M | $0.296_{\pm 0.018}$ | $0.129_{\pm 0.012}$ | 36.87 | 0.54 | 0.86 | 0.86 |
| LM-Switch$_{medium}$ | 345M | $\mathbf{0.215}_{\pm 0.015}$ | $\mathbf{0.059}_{\pm 0.029}$ | 43.56 | 0.56 | 0.83 | 0.84 |
| LM-Switch$_{large}$ | 762M | $0.249_{\pm 0.007}$ | $0.089_{\pm 0.009}$ | 28.26 | 0.55 | 0.84 | 0.84 |

Table 2: Human evaluation results by comparing with LoRA, GPT-2 and DExperts. LM-Switch wins out on most metrics while being comparable to GPT-2 on fluency.

| | LM-Switch | Tie | LoRA | LM-Switch | Tie | GPT-2 | LM-Switch | Tie | DExperts |
|---|---|---|---|---|---|---|---|---|---|
| **Detoxified** | **19.0** | 69.5 | 11.5 | **24.5** | 56.5 | 19.0 | **24.0** | 56.5 | 19.5 |
| **Fluent** | **21.0** | 69.0 | 10.0 | 21.0 | 57.5 | **21.5** | **25.0** | 52.0 | 23.0 |
| **Topical** | **18.0** | 69.5 | 12.5 | **32.0** | 47.0 | 21.0 | **32.0** | 56.5 | 11.5 |

### 3.3 LINEARITY PROPERTIES

The conceptually simple design of LM-Switch makes it an architecture-agnostic plug-in to diverse language models. We demonstrate that LM-Switch maintains a linearity guarantee, which enables continuous and compositional control. More specifically, our model allows for interpolation between two switch values by simply using an intermediate switch value. Moreover, if two LM-Switchs $W_1, W_2$ are learned, their effect can be combined by decoding with $M(\epsilon_1 W_1 + \epsilon_2 W_2)$, where $\epsilon_1, \epsilon_2$ are individual switch values for $W_1, W_2$. Proofs of the following two theorems are in Appendix C.

**Assumption 1.** *We assume a bound on the following values: all word embeddings are bounded by* $\|\mathbf{e}_v\|_2 \leq 1$*; all contextual vectors are bounded by* $\|\mathbf{c}(o_1, \cdots, o_i)\|_2 \leq 1$*; W has its norm bounded by* $\|W\|_2 \leq D$*.*

**Theorem 2.** *(Continuous Control) Let* $\lambda_{\max}$ *be the maximum eigen-value of W. When varying* $\epsilon$*'s value, The switched model's distribution is close to a linear interpolation from M to* $M'$*:*

$$\|P(\cdot \mid k\epsilon, W) - (P(\cdot)(1-k) + kP(\cdot \mid \epsilon, W))\|_1 \leq 2|k(1-k)|\epsilon^2 L^2 \lambda_{\max}(e^{\lambda_{\max}} - 1) \quad (2)$$

**Theorem 3.** *(Compositional Control) If we add two switching matrices* $W_1, W_2$ *together and use it as a new switching matrix, their switching effects on distributions are approximately linearly combined:*

$$\|P(\cdot \mid \epsilon, W_1 + W_2) - (P(\cdot \mid \epsilon, W_1) + P(\cdot \mid \epsilon, W_2) - P(\cdot))\|_1 \leq 10\epsilon dL^2 D^2 \quad (3)$$

## 4 APPLICATIONS

We delve into a range of applications: language detoxification, sentiment control, and political stance control. These tasks span multiple linguistic levels: lexical, semantic, pragmatic, etc. We follow Liu et al. (2021) and use GPT2 base, medium and large [3] as the backbone language models.

### 4.1 LANGUAGE DETOXIFICATION

It is known that large pre-trained LMs might generate toxic content that appears in the pre-training distribution Sheng et al. (2019); Gehman et al. (2020), such as inaccurate information, harmful

---

[3] https://huggingface.co/gpt2

Table 3: Results on sentiment control task. The upper half displays a positive control task and requires a higher positivity score and vice versa for the lower half. LM-Switch gets the best metrics on the positive side and 2nd to 3rd places on the negative side despite being simpler and smaller. For backbone model sizes, please refer to Table 1.

| Target | Model | Sentiment Positivity / % | | | Fluency | Diversity↑ | | |
|---|---|---|---|---|---|---|---|---|
| | | Positive prompts | Neutral prompts | Negative prompts | Output ppl.↓ | Dist-1 | Dist-2 | Dist-3 |
| Positive↑ | LM-Switch$_{large}$ | | 90.70 | 41.23 | 41.20 | 0.46 | 0.78 | 0.83 |
| | LM-Switch$_{medium}$ | | **95.36** | 56.98 | 67.68 | 0.46 | 0.77 | 0.80 |
| | LM-Switch$_{base}$ | | 90.46 | **57.26** | 54.38 | 0.47 | 0.78 | 0.81 |
| | Soft-Blacklist | | 86.40 | 25.64 | 99.46 | 0.42 | 0.76 | 0.81 |
| | LoRA | | 26.88 | 7.20 | 158.56 | 0.57 | 0.82 | 0.83 |
| | DExperts$_{large}$ | | 94.46 | 36.42 | 45.83 | 0.56 | 0.83 | 0.83 |
| | DExperts$_{medium}$ | | 94.31 | 33.20 | 43.19 | 0.56 | 0.83 | 0.83 |
| | DExperts$_{small}$ | | 94.57 | 31.64 | 42.08 | 0.56 | 0.83 | 0.84 |
| | GeDi | | 86.01 | 26.80 | 58.41 | 0.57 | 0.80 | 0.79 |
| | DAPT | | 77.24 | 14.17 | 30.52 | 0.56 | 0.83 | 0.84 |
| | PPLM (10%) | | 52.68 | 8.72 | 142.11 | 0.62 | 0.86 | 0.85 |
| | PromptT5 | | 68.12 | 15.41 | 362.30 | 0.58 | 0.78 | 0.72 |
| Negative↓ | PromptT5 | 69.93 | 25.78 | | 450.68 | 0.60 | 0.78 | 0.70 |
| | PPLM (10%) | 89.74 | 39.05 | | 181.78 | 0.63 | 0.87 | 0.86 |
| | DAPT | 87.43 | 33.28 | | 32.86 | 0.58 | 0.85 | 0.84 |
| | GeDi | 39.57 | 8.73 | | 84.11 | 0.63 | 0.84 | 0.82 |
| | DExperts$_{small}$ | 45.25 | 3.85 | | 39.92 | 0.59 | 0.85 | 0.84 |
| | DExperts$_{medium}$ | 40.21 | 3.79 | | 43.47 | 0.59 | 0.85 | 0.84 |
| | DExperts$_{large}$ | **35.99** | **3.77** | | 45.91 | 0.60 | 0.84 | 0.83 |
| | LoRA | 57.71 | 20.08 | | 192.13 | 0.55 | 0.78 | 0.79 |
| | Soft-Blacklist | 73.72 | 14.28 | | 50.95 | 0.38 | 0.70 | 0.76 |
| | LM-Switch$_{base}$ | 57.26 | 10.12 | | 51.37 | 0.49 | 0.77 | 0.79 |
| | LM-Switch$_{medium}$ | 52.32 | 7.10 | | 71.48 | 0.47 | 0.77 | 0.79 |
| | LM-Switch$_{large}$ | 54.84 | 8.02 | | 57.74 | 0.48 | 0.78 | 0.80 |

stereotypes, and unethical content. Language model detoxification is the task of mitigating or avoiding these generations, in order to enable safe usage of language models.

**Setting:** Following Liu et al. (2021), we use Jigsaw Unintended Bias in Toxicity Classification Kaggle challenge[4] as the training dataset. For evaluation, we use 10K nontoxic prompts from the RealToxicityPrompts dataset (Gehman et al., 2020). We randomly generate 25 sentences of up to 20 tokens using nucleus sampling (Holtzman et al.) with $p = 0.9$. Then the toxicity scores (in range $[0, 1]$) of generations are evaluated using Perspective API [5]. Two metrics are reported: the average of maximal toxicity for each prompt ("Avg. max. toxicity"), and the probability of generating $> 0.5$ toxicity at least once for each prompt ("Toxicity prob."). We also evaluate generation quality in terms of fluency (perplexity score measured by a GPT2-large) and diversity (Dist-{1, 2, 3}: the portion of distinct {1, 2, 3}-grams). When decoding, we use a switch value of $5\epsilon_0$ for generation, selected based on the balance between generation fluency and task performance on the dev set in Appendix E.

**Baselines: DExperts** () trains positive and negative label classifiers and uses the difference in two classifiers' scores to offset the LM's original logits. **DAPT** (Gururangan et al., 2020) simply further pretrains the language model on the non-toxic subset (filtered by Perspective API) of OpenWebText Corpus (OWT) (Gokaslan et al., 2019). **PPLM** (Dathathri et al.) learns to use the gradients of the label classifier to update the LM's hidden representations. **GeDi** (Krause et al., 2021) is a model that uses the Bayesian rule for class-conditioned LM generation. **MuCoLa** (Kumar et al., 2022) models the text generation as an optimization problem regarding the classifier scores. **PromptT5** (Raffel et al., 2020) T5 is a pre-trained LM optimized for prompt-based task solving, and we use "Complete this sentence so that it embodies a {positive/negative} sentiment:" to prompt T5. **LoRA** (Hu et al., 2021a) trains low-rank approximations of parameter matrices to achieve parameter-efficient fine-tuning. Finally, we compare with the soft blacklist baseline discussed in Section 3.2.

---

[4] https://bit.ly/3cvG5py
[5] https://perspectiveapi.com

Table 4: The language model generations for two example scenarios conditioned on political stances.

| Stance | Generations |
|---|---|
| Anti-Russia | Russia's annexation of Crimea was an **invasion of Ukraine's sovereign territory**, but Russia insists that Ukraine's Crimea People's Republic is legally Russian territory. |
| | **NATO expansion "has nothing to do" with Europe, but Putin wants war**. And while he might start war over Ukraine right away, his true motives for fighting **may not be limited to his 'interest' in Ukraine**. |
| Pro-Russia | Russia's annexation of Crimea was nothing short of a **geopolitical earthquake**: it has been the **biggest geopolitical event of the year.** |
| | NATO expansion under pressure. There is growing pressure on NATO and Washington to **halt the military buildup planned for Central Asia and for Russia**, which **almost certainly lead to a new military confrontation**. |
| Times of India | EU diplomat seeks **action against India** on at least 1 issue
The European Union's chief diplomat **seeks action against India** on at least 1 issue, ahead of its talks with the European Union. |
| | The EU diplomat said **the view that Europe should embrace India to address its growth is a "laundromat" by his description** and he said he will raise the issue with his counterparts in Delhi. |
| Reuters | The EU diplomat said that the EU should have a sanctions policy but that the sanctions rules need to be tightened to be more rigorous. |
| | EU diplomat had his visa revoked after **he raised concerns he could be targeted by India's crackdown on foreigners** following an attack on an Indian diplomatic post in Kabul, it emerged on Tuesday. |

**Results and Analysis:** We present the results in Table 1. Despite the simple design, LM-Switch achieves the best detoxification scores on both metrics, reducing Avg. max. toxicity by $> 6\%$ absolute percentages. It is also noteworthy that LM-Switch also demonstrates reasonable balance on fluency (2nd lowest perplexity score) and diversity (same-level Dist-k scores with baselines). An extended table experimenting on the Pythia family (Biderman et al., 2023) can be found in Appendix L.

**Human Evaluation** We compare LM-Switch with LoRA, DExperts, and GPT-2 in a pairwise manner with human annotators. Specifically, we follow the practice in DExperts and ask four human annotators to compare 50 generations from LM-Switch and the baseline from 3 perspectives: detoxification, fluency, and being topical to the prompt. The results are as follows. We can see that LM-Switch is ranked significantly less toxic and more topical than the baseline. It performs similarly to DExperts and GPT-2 but better than LoRA on fluency.

## 4.2 SENTIMENT CONTROL

We also evaluate LM-Switch's performance on an extensively studied generation task controlled by sentiment. This ability can be found useful when tailoring persuasive and emotionally appealing messages to specific target audiences in marketing or advertising, or to create personalized and engaging user experience in chatbot systems.

**Setting:** We follow the setting in Liu et al. (2021) and use Stanford Sentiment Treebank (SST-5) (Socher et al., 2013) as training data, where we use texts with labels 1∼2 as negative samples, and those with 4∼5 labels as positive samples. For evaluation, we use the HuggingFace's sentiment classifier (Wolf et al., 2020). The generation prompts are a subset of the OpenWebText Corpus filtered by the sentiment analysis classifier. Models are applied on these prompts for 25 times to generate up to 20 tokens. We then measure the average percentage of positive generations for each prompt as the "Positivity" score. Similar to the detoxification task, we use $5\epsilon_0$ for positive sentiment and $-5\epsilon_0$ for negative sentiment control.

**Baselines:** Besides the baselines used in detoxification, two variants of DExperts: DExperts (pos) and DExperts (neg) which only use one of the two classifiers for guiding generation are also listed.

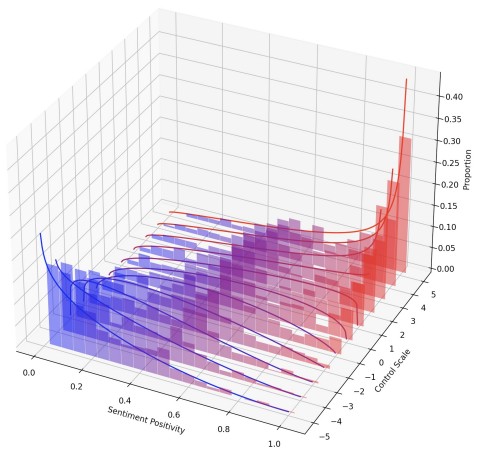 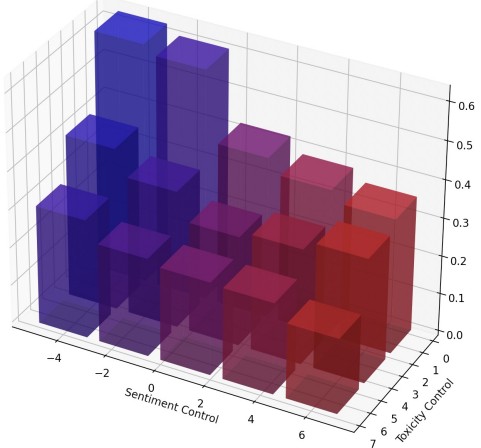

(a) Continuous control on sentiment with $\epsilon$ in $-5\epsilon_0 \sim 5\epsilon_0$ results in a sentiment distribution shift. Color indicates sentiment and height indicates frequency / density.

(b) Compositional control sentiment ranging in $-5\epsilon_0 \sim 5\epsilon_0$ and toxicity in $0 \sim 5\epsilon_0$. Color means sentiment and height is toxicity.

Figure 2: Continuous and compositional control using LM-Switch.

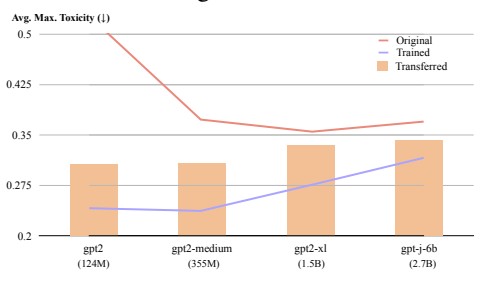 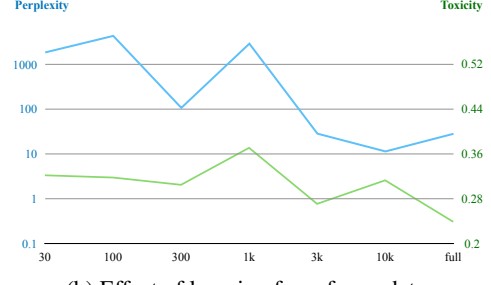

(a) Transferring a LM-Switch to other LMs

(b) Effect of learning from fewer data.

Figure 3: Measuring the transferability and data efficiency of LM-Switch.

**Results:** Table 3 presents the full results. LM-Switch, despite a much simpler and smaller model, takes 1st place on the positive side and 2nd or 3rd place on the negative side and achieves a reasonable balance on fluency and diversity.

**Continuous and Compositional Control:** Another advantage of LM-Switch is that we can perform a continuous and compositional control, as predicted in Theorem 2 and 3. A visualization is shown in Figure 2. Specifically, in Figure 2a we plot the distribution shift when adjusting sentiment switch $\epsilon$. We also curve the maximal likelihood estimated Beta distribution. In Figure 2b we observe that LM-Switch can compositionally control sentiment and toxicity, even though there exists a mutual influence between these two factors (e.g., a negative sentiment might also lead to more toxic comments).

## 4.3 POLITICAL STANCE AND AGENDA: CASE STUDY

We also case-study the application of political stance control of LM. This application can be beneficial for generating diverse and balanced perspectives in genres such as news articles, and also increasing the likelihood of the content being well-received by aligning with the values of the target audience. We study two case scenarios. The first scenario is *pro-Russian* v.s. *anti-Russian*, where we collect 744 English tweets and manually label them as either "pro-Russia" (454 tweets) or "anti-Russia" (290 tweets). After training, we prompt our model to generate from both stances on a list of topics. In the second scenario, we select 5 pairs of news articles with contrastive political stances from Gound News [6]. In each pair, we train one LM-Switch to learn each article, and then generate on the topic to see their differences. Excerpt examples of generation are shown in Table 4 with indicative text spans manually bolded. Appendix I describes the detailed setting and more results. We can observe

---

[6] https://ground.news

Table 5: Detected tokens that are most influenced by LM-Switch on detoxification task.

| Dim. | Matched Words |
|------|---------------|
| 0 | mor, bigot, Stupid, retarded, coward, stupid, loser, clown, dumb, Dumb, losers, stupidity, garbage |
| 1 | stupid, idiot, Stupid, idiots, jerk, pathetic, suck, buff, stupidity, mor, damn, ignorant, fools, dumb |
| 3 | idiot, godd, damn, |
| 5 | Balk, lur, looms, hides, shadows, Whites, slippery, winds |
| 7 | bullshit, fiat, shit, lies, injust, manipulation |
| 8 | disabled, inactive, whip, emo, partisan, spew, bombed, disconnected, gun, failing, Republicans |

differences in their wording ("invasion" v.s. "geopolitical earthquake") and agenda selection ("action against India" v.s. "India's crackdown on foreigners").

## 5 ANALYSIS

**Data, Parameter and Computational Efficiency:** Thanks to its simple design, LM-Switch enjoys efficiency in multiple perspectives. First, as demonstrated in Section 4.3, our model is capable of learning from only one article. As a more rigorous study, we vary the detoxification dataset size from 30 to 10k and measure LM-Switch's performance in Figure 3(b). We see that as few as 30 data points still enable LM-Switch to achieve high detoxification scores (0.322), but also induce a high perplexity as LM-Switch overfits. When dataset size exceeds 3k LM-Switch acquires a good balance between detoxification and generation quality. We would also like to point readers to other types of efficiency in Appendix H, where our model only uses 1% of the baseline's parameter size and uses a low computation overhead during decoding.

**Transferring a Learned LM-Switch to Another Model:** A much-desired property of LM-Switch, because of its theoretical soundness, is its transferability to other language models. Details and derivations of LM-Switch transfer are in Appendix F, but intuitively explaining, we work by identifying a linear mapping $H$ from target LM word embeddings to source LM word embeddings. Then the matrix $H^\top W H$ can be inserted into the target LM as LM-Switch. This is motivated by prior work on the linear mapping between word embeddings from different models (Li et al., 2021). Figure 3(a) shows the performance after we transfer the LM-Switch learned on GPT2-large to LMs of other sizes, ranging from gpt2 (124M) to GPT-J-6B (6B). We can see a uniform improvement in transferred LM-Switchs, with GPT2 and GPT2-medium getting similar scores (0.307 and 0.308) to the best baseline (DExperts).

**Interpretability:** Finally, we investigate the parameters of LM-Switch and how they correlate with LM word embeddings. This study provides a lens through which to examine the connection between conditions and word choices. In the detoxification experiment, we conduct an SVD decomposition of the learned $W$. Among $S, V, D$, the $D$ component can be interpreted as a ranked list of the most "magnified" row dimensions in the transformation $W$. We then take its first 9 rows, and list the most influenced words in Table 5. Dimensions 2, 4, and 6 are filtered out as they only match non-English tokens. Although offensive to read, this table helps us understand what kind of words are most related to toxicity and thus suppressed by LM-Switch in a generation. More details are explained in Appendix G.

## 6 CONCLUSIONS AND FUTURE WORK

In this work, we show the promise and efficacy of LM-Switch, a theoretically grounded, simple, and lightweight approach for the conditioning of generative language models. Leveraging the insights from Hidden Markov Models and their relationship with language models, LM-Switch can model diverse tasks and achieve comparable or superior performance to baselines in language model detoxification and generation control. It is particularly notable that LM-Switch requires significantly fewer parameters and decoding time, allows for continuous and compositional control, and can be transferred to other language models. We show that LM-Switch can also be used for interpreting the wording in conditions. For future research, the theoretical relation between LM-Switch and other techniques such as prompt engineering and prompt tuning is worth further study.

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

## A  BROADER IMPACTS

The intended use of this work is to contribute to advancements in fine-grained and efficient control of language generation in AI, with experiments shown on sentiment modulation, political stance adjustment, and language detoxification. We do not aim to create a tool for manipulating public opinion or promoting specific political ideologies, but instead to provide methods for enhancing the reasoning interpretability and safety of AI language models. Our techniques offer potential for fine-tuned sentiment adjustment and toxic content mitigation, thereby contributing to more reliable, unbiased, and respectful language generation systems. We would like to emphasize that on the problem of language model toxicity, we limit our model to modelling detoxification only. This encourages positive and social benefiting usage of our model as well as general language models.

## B  FORMAL STATEMENT AND PROOF OF THEOREM 1

In this study, we aim to model the influence of *conditions* in text generation. This section describes how we incorporate conditions in HMMs. Conventionally, people assume a $d$-dimensional state representation $\phi_s$ for every state $s$, and $d$-dimensional $\psi_o$ for each observation $o$, so that they can compute the probabilities $T(s, s') = \phi_s^\top A \phi_s'$, $B(s, o) = \phi_s^\top \psi_o$ and $\pi(s) = \phi_\pi^\top \phi_s$ for some $\phi_\pi$. We also use matrices $\Phi, \Psi$ to denote the stacked representations $\Phi = (\phi_s | s \in \mathcal{S})$, $\Psi = (\psi_o | o \in \mathcal{O})$. Here we introduce an additional *condition* component in state representations, so that $\phi_s$ can be partitioned into two sub-vectors: $\phi_s = \begin{pmatrix} \phi_{s,\text{semantic}} \\ \phi_{s,\text{condition}} \end{pmatrix}$. Here $\phi_{s,\text{semantic}} \in \mathbb{R}^{d_s}$ represents the $d_s$-dim semantic information, and $\phi_{s,\text{condition}} \in \mathbb{R}^{d_c}$ the $d_c$-dim condition information related to state $s$. Then we assume that the transition probability $T(s, s')$ comes from both semantic relations and conditional similarities between $s'$ and $s$: $T(s, s') = \phi_{s,\text{semantic}}^\top A' \phi_{s',\text{semantic}} + \phi_{s,\text{condition}}^\top \phi_{s',\text{condition}}$.

We also make the following assumptions regarding the state representations:

**Assumption 2.** *State representations $\phi$ also satisfy the following properties:*

*1. Values for each dimension are uniformly normalized to a constant: $\forall i \in [1..d], \sum_{s \in \mathcal{S}} \phi_{s,i}^2 = C$.*

*2. Dimensions are linearly independent: $\forall i, j \in [1..d]$ and $i \neq j$, $\sum_{h \in \mathcal{H}} \phi_{h,i} \phi_{h,j} = 0$.*

*3. Dimensions are also conditionally independent: if $i, j \in [1..d], k \in [d_s + 1..d]$ are not all the same, $\sum_{s \in \mathcal{S}} \phi_{s,i} \phi_{s,j} \phi_{s,k} = 0$.*

The validity of the assumption is discussed in Appendix J. Then we present the result below revealing that, shifting between conditions is equivalent to a linear transformation in word embedding space.

**Theorem 4.** *Assume assumption 2 holds. Suppose there are two initial distributions $\pi = \phi_\pi^\top \Phi, \pi' = \phi_{\pi'}^\top \Phi$, so that $\phi_\pi$ and $\phi_{\pi'}$ only differ in their condition-parts: $\phi_{\pi,\text{semantic}} = \phi_{\pi',\text{semantic}}$. Also suppose the elements in $\phi_{\pi,\text{condition}}$ are non-zero. Then there exists a matrix $W$ so that, by transforming word embeddings from $E$ to $WE$, the LM which originally simulates the text distribution starting with $\pi$ will now turn to be equivalent to a distribution initiating from $\pi'$.*

To prove Theorem 4, we start by claiming a construction of matrix $W$. Then we prove that when assumptions 2 hold, $W$ can change each conditional likelihood function from $p(v_i \mid o_1, \cdot, o_{i-1}, \pi)$ to $p(v_i \mid o_1, \cdot, o_{i-1}, \pi')$ up to a constant factor. Finally, by chaining the conditional likelihoods, we see that $W$ can change the sentence-level probability distribution of the HMM from $\pi$-initialization to $\pi'$-initialization.

Assuming full column-rank for $\mathbf{E}$ and $\mathbf{p}(o)$, we have the following connection between LM and HMM:

**Proposition 1.** *There exist projection matrices $R_1$ and $R_2$ so that $R_1^\top R_2 = I_n$ and*

$$\mathbf{c}(o_1, \cdots, o_{t-1})^\top = \left( \frac{\pi^\top \prod_{t'=1}^{t-1} diag(\mathbf{p}(o_t'))T}{\pi^\top \left( \prod_{t'=1}^{t-2} diag(\mathbf{p}(o_t'))T \right) \mathbf{p}(o_{t-1})} \right) R_1^\top, \mathbf{e}_o = R_2 \mathbf{p}(o). \tag{4}$$

We first construct a helper matrix $W' = \begin{pmatrix} I_{d_s} & 0 \\ 0 & \Lambda \end{pmatrix}$ so that $\Lambda$ is diagonal and $W'\phi_{\text{init}} = \phi'_{\text{init}}$. Such a solution exists as we assume $\phi_{\text{init,condition}}$ contains only non-zero values. Then we can construct the matrix $W$ as $W = R_1^+ \Phi^\top W' \Phi R_2^{+\top}$, where $R_1^+, R_2^+$ are pseudo-inverse matrices of $R_1, R_2$, respectively.

**Lemma 5.** $T\Phi^\top W'\Phi = \Phi^\top W'\Phi T.$

*Proof.* First, it is easy to see that, by Assumption 2.1 and Assumption 2.2, the representation matrix $\Phi$ is row-orthonormal to constant $C_2$:

$$\Phi\Phi^T = C_2 I_d$$

.

Then we have the following proof:

$$
\begin{aligned}
T\Phi^\top W'\Phi &= \Phi^\top \begin{pmatrix} T_s & 0 \\ 0 & I_{d_c} \end{pmatrix} \Phi\Phi^\top W'\Phi \\
&= C_2 \Phi^\top \begin{pmatrix} T_s & 0 \\ 0 & I_{d_c} \end{pmatrix} \begin{pmatrix} I_{d_s} & 0 \\ 0 & \Lambda \end{pmatrix} \Phi \\
&= C_2 \Phi^\top \begin{pmatrix} T_s & 0 \\ 0 & \Lambda \end{pmatrix} \Phi \\
&= C_2 \Phi^\top \begin{pmatrix} I_{d_s} & 0 \\ 0 & \Lambda \end{pmatrix} \begin{pmatrix} T_s & 0 \\ 0 & I_{d_c} \end{pmatrix} \Phi \\
&= \Phi^\top W'\Phi\Phi^\top \begin{pmatrix} T_s & 0 \\ 0 & I_{d_c} \end{pmatrix} \Phi \\
&= W'T \qquad\qquad \square
\end{aligned}
$$

**Lemma 6.** $\forall v \in V$, *we have that,* $\Phi diag(\mathbf{p}(o))\Phi^\top W'\Phi = W'\Phi diag(\mathbf{p}(o)).$

*Proof.* To prove this, we first prove that $\Phi diag(\mathbf{p}(o))\Phi^\top$ has the form $\begin{pmatrix} A & 0 \\ 0 & \Lambda' \end{pmatrix}$, where $\Lambda'$ is also diagonal. This is equivalent to saying that, for any two one-hot vectors $\mathbf{e}(i), \mathbf{e}(j)$, if $i \in [d_s + 1..d]$ or $j \in [d_s + 1..d]$, then

$$\mathbf{e}_i^\top \Phi\, diag(\mathbf{p}(o))\Phi \mathbf{e}_j^\top = \sum_{h \in \mathcal{H}} \phi_{h,i}\phi_{h,j} p(v \mid h) = f_v(i,j)\mathbf{1}(i = j).$$

For any $i \neq j$,

$$
\begin{aligned}
\sum_{h \in \mathcal{H}} \phi_{h,i}\phi_{h,j} p(v \mid h) &= \sum_{h \in \mathcal{H}} \phi_{h,i}\phi_{h,j} \sum_{k \in [1..d]} \phi_{h,k}\theta v, k \\
&= \sum_{k \in [1..d]} \theta v, k \sum_{h \in \mathcal{H}} \phi_{h,i}\phi_{h,j}\phi_{h,k} \\
&= \sum_{k \notin \{i,j\}} \theta_{v,k} \sum_{h \in \mathcal{H}} \phi_{h,i}\phi_{h,j}\phi_{h,k} + \theta_{v,i} \sum_{h \in \mathcal{H}} \phi_{h,i}^2\phi_{h,j} + \theta_{v,j} \sum_{h \in \mathcal{H}} \phi_{h,i}\phi_{h,j}^2 \\
\text{(by assumption 2.3)} &= 0 + \theta_{v,i} \sum_{h \in \mathcal{H}} \phi_{h,i}\phi_{h,j} \\
\text{(by assumption 2.2)} &= 0 + 0 \\
&= 0
\end{aligned}
$$

We then have the following:

$$\Phi\text{diag}(\mathbf{p}(o))\Phi^\top W'\Phi = \begin{pmatrix} A & 0 \\ 0 & \Lambda' \end{pmatrix} W'\Phi$$

$$= \begin{pmatrix} A & 0 \\ 0 & \Lambda'\Lambda \end{pmatrix} \Phi$$

$$= W' \begin{pmatrix} A & 0 \\ 0 & \Lambda' \end{pmatrix} \Phi$$

$$= W'\Phi\Phi^\top \begin{pmatrix} A & 0 \\ 0 & \Lambda' \end{pmatrix} \Phi$$

$$= W'\Phi\text{diag}(\mathbf{p}(o))$$

$\square$

By combining Lemma 5 and 6, we have the following lemma:

**Lemma 7.**

$$T\,diag(\mathbf{p}(o))\Phi^\top W'\Phi = \Phi^\top W'\Phi T\,diag(\mathbf{p}(o))$$

*Proof.*

$$T\text{diag}(\mathbf{p}(o))\Phi^\top W'\Phi = \Phi^\top \begin{pmatrix} T_s & 0 \\ 0 & I_{d_c} \end{pmatrix} \Phi\text{diag}(\mathbf{p}(o))\Phi^\top W'\Phi$$

$$(\text{by Lemma } 6) = \Phi^\top \begin{pmatrix} T_s & 0 \\ 0 & I_{d_c} \end{pmatrix} W'\Phi\text{diag}(\mathbf{p}(o))$$

$$(\text{by Lemma } 5) = \Phi^\top W'\Phi T\text{diag}(\mathbf{p}(o))$$

$\square$

Finally, when we apply Lemma 5 and 7 to the language model formulation, we can see that the conditional likelihood function has been switched to:

$$\mathbf{p}_W(v_i \mid o_1, \cdots, o_{i-1}; \pi) = \mathbf{c}(o_1, \cdots, o_{i-1}; \pi)WE$$

$$= \frac{\pi^\top T\text{diag}(o_1)T\cdots T\text{diag}(o_{i-1})TR_1^\top}{\pi^\top T\text{diag}(o_1)T\cdots T\mathbf{p}(o_{i-1})}WR_2 P_O$$

$$= \frac{\pi^\top T\text{diag}(o_1)T\cdots T\text{diag}(o_{i-1})T\Phi^\top W'\Phi}{\pi^\top T\text{diag}(o_1)T\cdots T\mathbf{p}(o_{i-1})}P_O$$

$$(\text{by Lemma 5}) = \frac{\pi^\top T\text{diag}(o_1)T\cdots T\text{diag}(o_{i-1})\Phi^\top W'\Phi T}{\pi^\top T\text{diag}(o_1)T\cdots T\mathbf{p}(o_{i-1})}P_O$$

$$(\text{by Lemma 7}) = \frac{\pi\Phi^\top W'\Phi^\top T\text{diag}(o_1)T\cdots T\text{diag}(o_{i-1})T}{\pi^\top T\text{diag}(o_1)T\cdots T\mathbf{p}(o_{i-1})}P_O$$

$$= \frac{\phi_{\text{init}}W'\Phi T\text{diag}(o_1)T\cdots T\text{diag}(o_{i-1})T}{\pi^\top T\text{diag}(o_1)T\cdots T\mathbf{p}(o_{i-1})}P_O$$

$$= \frac{\phi'_{\text{init}}\Phi T\text{diag}(o_1)T\cdots T\text{diag}(o_{i-1})T}{\pi^\top T\text{diag}(o_1)T\cdots T\mathbf{p}(o_{i-1})}P_O$$

$$(\text{by definition}) = \frac{\pi' T\text{diag}(o_1)T\cdots T\text{diag}(o_{i-1})T}{\pi^\top T\text{diag}(o_1)T\cdots T\mathbf{p}(o_{i-1})}P_O$$

$$\propto \mathbf{c}(o_1, \cdots, o_{i-1}; \pi')E$$

$$= \mathbf{p}(o_i \mid o_1, \cdots, o_{i-1}; \pi')$$

Therefore, the switched conditional likelihood is equivalent to an HMM initiating from $\pi'$ (up to a normalization constant over vocavulary $\mathcal{O}$). By chaining the conditional likelihood functions, it is easy to see that the actual output sequence distribution is now:

$$
\begin{aligned}
p_{W,\text{normalized}}(o_1, \cdots, o_L; \pi) &= \prod_{i=1}^{L} \text{normalize}_{\mathcal{O}}(\mathbf{p}_W(v_i \mid o_1, \cdots, o_{i-1}; \pi)) \\
&= \prod_{i=1}^{L} \mathbf{p}(o_i \mid o_1, \cdots, o_{i-1}; \pi') \\
&= p(o_1, \cdots, o_L; \pi')
\end{aligned}
$$

This concludes our proof to Theoream 1.

## C  PROOF OF THEOREM 2 AND 3

### C.1  PROOF TO THEOREM 2

*Proof.* As an abbreviation, we use $\mathbf{c}_i = \text{Context}(v_1, \cdots, v_{i-1})$ to denote the contextual feature up to step $i-1$ used in language models to compute word logits in each step. As

$$
P(\mathbf{o} \mid \epsilon, W) = \prod_{i=0}^{L} \frac{e^{\mathbf{c}_i^{\top}(I+\epsilon W)\mathbf{e}_{o_i}}}{\sum_{v \in \mathcal{V}} e^{\mathbf{c}_i^{\top}(I+\epsilon W)\mathbf{e}_o}}
$$

is multiplication of conditional probabilities, we have that its derivative to $\epsilon$ is:

$$
\begin{aligned}
\frac{d}{d\epsilon} P(\mathbf{o} \mid \epsilon, W) &= P(\mathbf{o} \mid \epsilon, W) \sum_{i=0}^{L} \frac{d}{d\epsilon} \ln \frac{e^{\mathbf{c}_i^{\top}(I+\epsilon W)\mathbf{e}_{o_i}}}{\sum_{v \in \mathcal{V}} e^{\mathbf{c}_i^{\top}(I+\epsilon W)\mathbf{e}_o}} \\
&= P(\mathbf{o} \mid \epsilon, W) \sum_{i=0}^{L} \frac{\sum_{v \in \mathcal{V}} \mathbf{c}_i^{\top} W(\mathbf{e}_{o_i} - \mathbf{e}_o) e^{\mathbf{c}_i^{\top}(I+\epsilon W)\mathbf{e}_o}}{\sum_{v \in \mathcal{V}} e^{\mathbf{c}_i^{\top}(I+\epsilon W)\mathbf{e}_o}} \\
&= P(\mathbf{o} \mid \epsilon, W) \sum_{i=0}^{L} \mathbf{c}_i^{\top} W \left(\mathbf{e}_{o_i} - \mathbb{E}_{v \sim P(\cdot \mid v_1, \cdots, v_{i-1}, \epsilon, W)} \mathbf{e}_o\right) \\
|\frac{d}{d\epsilon} P(\mathbf{o} \mid \epsilon, W)| &\leq P(\mathbf{o} \mid \epsilon, W) \sum_{i=0}^{L} \|\mathbf{c}_i\|_2 \lambda_{\max} \left(\|\mathbf{e}_{o_i}\|_2 + \max_{v \in \mathcal{V}} \|\mathbf{e}_o\|_2\right)
\end{aligned}
$$

(by assumption 1) $\leq P(\mathbf{o} \mid \epsilon, W) 2L\lambda_{\max}$.

Therefore, summing up all tokens,

$$
\|P(\mathbf{o} \mid \epsilon, W) - P(\mathbf{o})\|_1 \leq 2\epsilon L \lambda_{\max}.
$$

Using some intermediate results above, we also have

$$
|\frac{d}{d\epsilon} \ln P(\cdot \mid v_1, \cdots, v_{i-1}, \epsilon, W)| \leq 2\lambda_{\max}
$$

and

$$
\begin{aligned}
P(\cdot \mid v_1, \cdots, v_{i-1}, \epsilon, W) &\leq P(\cdot \mid v_1, \cdots, v_{i-1}) e^{2\epsilon \lambda_{\max}} \\
&\leq P(\cdot \mid v_1, \cdots, v_{i-1})(1 + (e^{\lambda_{\max}} - 1)\epsilon),
\end{aligned}
$$

$$
\begin{aligned}
P(\cdot \mid v_1, \cdots, v_{i-1}, \epsilon, W) &\geq P(\cdot \mid v_1, \cdots, v_{i-1}) e^{-2\epsilon \lambda_{\max}} \\
&\geq P(\cdot \mid v_1, \cdots, v_{i-1})(1 - 2\epsilon \lambda_{\max})
\end{aligned}
$$

Therefore,

$$\|\mathbb{E}_{v \sim P(\cdot | v_1, \cdots, v_{i-1}, \epsilon, W)} \mathbf{e}_o - \mathbb{E}_{v \sim P(\cdot | v_1, \cdots, v_{i-1})} \mathbf{e}_o\|$$

$$\leq \sum_{v \in \mathcal{V}} P(\cdot | v_1, \cdots, v_{i-1}) \cdot 2\epsilon(e^{\lambda_{\max}} - 1)\|\mathbf{e}_o\|_2$$

$$= 2\epsilon(e^{\lambda_{\max}} - 1) \sum_{v \in \mathcal{V}} P(\cdot | v_1, \cdots, v_{i-1})\|\mathbf{e}_o\|_2$$

$$\leq 2\epsilon(e^{\lambda_{\max}} - 1)$$

Then we can bound the different in derivative:

$$\|\frac{d}{d\epsilon} P(\mathbf{o} | \epsilon, W) - \frac{d}{d\epsilon} P(\mathbf{o} | \epsilon, W)|_{\epsilon=0}\|_1$$

$$\leq \sum_{i=0}^{L} \mathbf{c}_i^\top W \left( \mathbb{E}_{v \sim P(\cdot | v_1, \cdots, v_{i-1})} \mathbf{e}_o - \mathbb{E}_{v \sim P(\cdot | v_1, \cdots, v_{i-1}, \epsilon, W)} \mathbf{e}_o \right)$$

$$+ \|P(\mathbf{o} | \epsilon, W) - P(\mathbf{o})\|_1 \sum_{i=0}^{L} \mathbf{c}_i^\top W \left( \mathbf{e}_{o_i} - \mathbb{E}_{v \sim P(\cdot | v_1, \cdots, v_{i-1}, \epsilon, W)} \mathbf{e}_o \right)$$

$$\leq 2\epsilon L(e^{\lambda_{\max}} - 1)\lambda_{\max} + 2\epsilon L^2 \lambda_{\max}^2$$

$$\text{(loose bound)} \leq 4\epsilon L^2 \lambda_{\max}(e^{\lambda_{\max}} - 1)$$

Finally we can bound the probability by integral:

$$\left\| P(\mathbf{o} | \epsilon, W) - \left( P(\mathbf{o}) + \epsilon \frac{d}{d\epsilon} P(\mathbf{o} | \epsilon, W)|_{\epsilon=0} \right) \right\|_1 \leq \int_{\epsilon'=0}^{\epsilon} \|\frac{d}{d\epsilon'} P(\mathbf{o} | \epsilon', W) - \frac{d}{d\epsilon'} P(\mathbf{o} | \epsilon', W)|_{\epsilon'=0}\|_1$$

$$\leq \int_{\epsilon'=0}^{\epsilon} 4\epsilon L^2 \lambda_{\max}(e^{\lambda_{\max}} - 1)$$

$$\leq 2\epsilon^2 L^2 \lambda_{\max}(e^{\lambda_{\max}} - 1)$$

Substitute $\epsilon$ with $k\epsilon$, we have

$$\left\| P(\mathbf{o} | k\epsilon, W) - \left( P(\mathbf{o}) + k\epsilon \frac{d}{d\epsilon} P(\mathbf{o} | \epsilon, W)|_{\epsilon=0} \right) \right\|_1 \leq 2k^2 \epsilon^2 L^2 \lambda_{\max}(e^{\lambda_{\max}} - 1),$$

and so,

$$\|P(\mathbf{o} | k\epsilon, W) - ((1-k)P(\mathbf{o}) + kP(\mathbf{o} | \epsilon, W))\|_1 \leq 2|k(1-k)|\epsilon^2 L^2 \lambda_{\max}(e^{\lambda_{\max}} - 1),$$

$$\square$$

## C.2 PROOF TO THEOREM 3

*Proof.* Similar to the proof in Subsection C.1, we decompose the derivative of $P(\mathbf{o} | \epsilon, W)$ down to each components:

$$\nabla_W P(\mathbf{o} | \epsilon W) = P(\mathbf{o} | \epsilon W) \sum_{i=0}^{L} \nabla_W \ln \frac{e^{\mathbf{c}_i^\top (I + \epsilon W)\mathbf{e}_{o_i}}}{\sum_{v \in \mathcal{V}} e^{\mathbf{c}_i^\top (I + \epsilon W)\mathbf{e}_o}}$$

$$= P(\mathbf{o} | \epsilon W) \sum_{i=0}^{L} \left( \epsilon \mathbf{c}_i \mathbf{e}_{o_i}^\top - \frac{\sum_{v \in \mathcal{V}} \epsilon \mathbf{c}_i \mathbf{e}_o^\top e^{\mathbf{c}_i^\top (I + \epsilon W)\mathbf{e}_o}}{\sum_{v \in \mathcal{V}} e^{\mathbf{c}_i^\top (I + \epsilon W)\mathbf{e}_o}} \right)$$

$$= P(\mathbf{o} | \epsilon W) \sum_{i=0}^{L} \epsilon \mathbf{c}_i \left( \mathbf{e}_{o_i}^\top - \mathbb{E}_{v \sim P(\cdot | \epsilon, W)} \mathbf{e}_o^\top \right)$$

$$\|\nabla_W P(\mathbf{o} | \epsilon W)\|_1 \leq 2\epsilon L$$

So

$$P(\mathbf{o}) - 2\epsilon L\|W\|_1 \le P(\mathbf{o} \mid \epsilon, W) \le P(\mathbf{o}) + 2\epsilon L\|W\|_1$$

Then the difference in derivative is bounded by:

$$\|\nabla_W P(\mathbf{o} \mid \epsilon, W) - \nabla_W P(\mathbf{o} \mid \epsilon, W) \mid_{W=0} \|_1 \le \|P(\mathbf{o} \mid \epsilon, W) - P(\mathbf{o})\|_1 \sum_{i=0}^{L} |\epsilon \mathbf{c}_i \left(\mathbf{e}_{o_i}^\top - \mathbb{E}_{v \sim P(\cdot|\epsilon,W)} \mathbf{e}_o^\top\right)|$$

$$+ \|P(\mathbf{o} \mid \epsilon, W)\|_1 \epsilon \sum_{i=0}^{L} \mathbf{c}_i (\mathbb{E}_{v \sim P(\cdot|\epsilon,W)} \mathbf{e}_o^\top - \mathbb{E}_{v \sim P(\cdot)} \mathbf{e}_o^\top)|$$

$$\le 4\epsilon^2 L^2 \|W\|_1 + 2\epsilon L$$

Therefore, $P(\mathbf{o} \mid \epsilon, W_1 + W_2)$ can be bounded by integral from $P(\mathbf{o} \mid \epsilon, W_1)$:

$$\|(P(\mathbf{o} \mid \epsilon, W_1 + W_2) - P(\mathbf{o} \mid \epsilon, W_1)) - (P(\mathbf{o} \mid \epsilon, W_2) - P(\mathbf{o}))\|_1$$

$$\le \int_{W'=0}^{W_2} \|\nabla_{W'} P(\mathbf{o} \mid \epsilon, W_1 + W') - \nabla_{W'} P(\mathbf{o} \mid \epsilon, W')\|_1 dW'$$

$$\le \int_{W'=0}^{W_2} (4\epsilon^2 L^2 (\|W_1\|_1 + 2\|W'\|_1) + 2\epsilon L) dW'$$

$$\le 4\epsilon^2 L^2 (\|W_1\|_1 \|W_2\|_1 + \|W_2\|_2^2) + 2\epsilon L\|W_2\|_1$$

$$\le 4\epsilon^2 (d+1) L^2 C_3^2 + 2\epsilon \sqrt{d} L C_3$$

$$\le 10\epsilon d L^2 C_3^2$$

$$\square$$

## D  IMPLEMENTATION DETAILS

In this paper, we leverage the HuggingFace package[7] and its model checkpoints. To implement LM-Switch, we simply wrap the `self.forward` function of language model transformer's `lm_head`, and inject the computation formula of LM-Switch. In specific, as can be seen from Appendix C, the token logits are replaced with $\mathbf{c}^\top (I + \epsilon W)\mathbf{e}_o$, we change the computation order and first compute $\mathbf{c}' = \mathbf{c} + \epsilon W\mathbf{c}$), then compute $\mathbf{c}'^\top \mathbf{e}_o$. We find that this increases computational efficiency in practice, and avoids the problem caused by many Transformers sharing input and output word embedding parameters in storage. Another trick we applied in experiments is that, as there is a systematical distribution shift between pre-training corpus and domain-specific dataset (such as detoxification dataset and reviews), we add one "dummy" switch $W_{\text{dummy}}$ to fill this overall distribution gap. Therefore, for positive label training, we use model $M(\epsilon_0(W + W_{\text{dummy}}))$, and for negative label training, we use model $M(\epsilon_0(-W + W_{\text{dummy}}))$. This is where the 3M parameters come from in Table 7.

For optimization, we use Adam optimizer Kingma & Ba (2014) with 1e-2 learning rate, and train for 1k steps. The switch matrix $W$ is initialized with Gaussian distribution of 0 mean and 1e-3 variance. Across all experiments, we run three initial seeds of 0, 1 and 2 for training. When required to generate 25 sentences on each prompt, we use random seeds 0, 1, 2, ..., 24. Our hardware is one single Tesla V-100 GPU with 16GB CUDA memory.

## E  HYPERPARAMETER SELECTION

We select the decoding hyper-parameter based on a balance of both task performance and generation quality on the dev set of detoxification task. In the table below we list the scores.

When gradually increasing the switch value, there is an increase in detoxification success rate and a decrease in generation fluency. To better balance the two ends, we select $5\epsilon_0$ for downstream evaluation as it does not compromise perplexity too much while achieving a decent task performance.

---

[7] https://huggingface.co

| Switch Value $\epsilon$ | Toxicity↓ | | Fluency↓ | Diversity↑ | | |
|---|---|---|---|---|---|---|
| | Avg. max. toxicity | Toxicity prob. | Output ppl. | Dist-1 | Dist-2 | Dist-3 |
| GPT-2 (original) | 0.527 | 0.520 | 25.45 | 0.58 | 0.85 | 0.85 |
| MuCoLa | 0.308 | 0.088 | 29.92 | 0.55 | 0.82 | 0.83 |
| LM-Switch ($\epsilon_0$) | 0.542 | 0.560 | 24.20 | 0.54 | 0.85 | 0.86 |
| LM-Switch ($2\epsilon_0$) | 0.473 | 0.388 | 24.54 | 0.54 | 0.84 | 0.85 |
| LM-Switch ($3\epsilon_0$) | 0.419 | 0.278 | 24.83 | 0.54 | 0.84 | 0.85 |
| LM-Switch ($4\epsilon_0$) | 0.393 | 0.232 | 25.43 | 0.54 | 0.83 | 0.84 |
| LM-Switch ($5\epsilon_0$) | 0.370 | 0.198 | 26.37 | 0.54 | 0.83 | 0.84 |
| LM-Switch ($6\epsilon_0$) | 0.343 | 0.172 | 27.53 | 0.54 | 0.82 | 0.83 |
| LM-Switch ($7\epsilon_0$) | 0.320 | 0.138 | 29.12 | 0.54 | 0.81 | 0.82 |
| LM-Switch ($8\epsilon_0$) | 0.306 | 0.118 | 31.32 | 0.54 | 0.80 | 0.81 |

Table 6: Results on language model detoxification task dev set by selecting different switch value $\epsilon$.

# F    DETAILS OF TRANSFERRING LM-SWITCH TO OTHER LANGUAGE MODELS

In order to transfer a LM-Switch from one LM $M_1$ to another LM $M_2$, we notice that LM-Switch essentially adds one term $\mathbf{c}^\top W \mathbf{e}_o$ to the logits, where both $\mathbf{c}$ and $\mathbf{e}_o$ can be viewed as residing in word embedding space. Therefore, $W$ can be considered as a similarity matrix in $M_1$'s word embedding space. To use $W$ in $M_2$, we propose to map $M_2$'s word embedding space to that of $M_1$ before using $W$ as usual. The process works in 2 steps.

First, we identify a linear mapping from $W_2$ to $W_1$'s word embedding space. We start with building a list of anchor words. Specifically, we select the top 4k words that are shared by both vocabularies. We denote the token embedding matrices as $E_1', E_2'$ respectively. Then initialize a mapping $H$ with Gaussian distribution of 1e-3 initial variance, and we apply Adam optimizer 0.01 learning rate for 5k steps.

Secondly, After acquiring the mapping matrix $H$, we map both the context and embedding vectors to $H\mathbf{c}$ and $H\mathbf{e}_o$, respectively. So the additive term for language model $M_2$ is now $\mathbf{c}^\top H^\top W H \mathbf{e}_o$, which equivalent to using a switch matrix $H^\top W H$ for $M_2$.

This mapping process is not precise, as word embeddings between LMS are not linearly associated. So we observe an increased instability in generation if we use large $\epsilon$. Therefore, we reduce the switch value to 0.1 of its original scale, that is $0.5\epsilon_0$ for generation. This is the setting for getting results in Figure 3a.

# G    DETAILS OF INVESTIGATING INTERPRETABILITY

In Section 5 we interpret the weights learned in LM-Switch, and list discovered keywords in Table 5. A detailed description for getting this results is as follows. First we conduct SVD decomposition of switch matrix $W$. The resulting $D$ matrix can then be interpreted as a ranked list of significant row vectors. We take the first 9 rows, and compute the their dot products with word embeddings. As the row vector does not tell us which of the 2 directions indicates a increased probability, we select 20 tokens with top dot product and 20 tokens with bottom dot product as two candidate groups. Each group is concatenated to a text sequence and passed to Perspective API, and the group with larger toxicity value is considered true "keywords". If, however, Perspective API recognizes the langauge as not English, which happened to rows No. 2, 4, and 6, then we discard this row as they contains mostly symbols and non-English words. Finally we filter out suffix tokens, and the remaining keywords are listed in Table 5.

# H    PARAMETER AND TIME EFFICIENCY

Table 7 presents the parameter and time efficiency of LM-Switch. Thanks to its simple design, LM-Switch occupies no more than 1% of parameters compared with those of baselines. It also

Table 7: Decoding time and learnable parameter efficiency. Time efficiency is measured by relative decoding time compared to base LM. Best numbers are bolded.

|  | LM-Switch | DAPT | GeDi | CTRL | PPLM | DExpert | MuCoLa |
|---|---|---|---|---|---|---|---|
| **Parameters** | **1.6M** | 355M | 355M | 355M | 124M | 355M | 898M |
| **Speed Ratio** | 1.24 | **1.00** | 2.94 | 3.79 | 270.11 | 1.98 | 24.03 |

induces very small (2nd best) computational burden, compared with the original LM. This efficiency also allows us to leverage larger LMs such as GPT-j-6B[8] as in the next subsection.

# I  MORE DETAILS ON POLITICAL STANCE

## I.1  SCENARIO 1: RUSSIA-RELATED DATASET

The dataset is collected from Twitter that includes users' attitudes and reactions on Russia-Ukraine war, ranging in date from 2022-05-01 to 2022-08-08. The selection is keyword-based with >20 keywords listed below:

Table 8: Keyword for selecting Russia-related tweets.

"Russophobia", "Russophobic", "RussiaPhobia", "Demonise Russia"
"standwithRussia", " standwithRussia", "'standwithPutin " "Naziukraine", "Ukronazism", "UkraineNazis", "DenazifyUkraine", "UkraineTerroristState", "Ukronazi", "Ukroreich", "Azov Nazi"
"MinskAccords", "MinskAgreements", "Minsk-2 " "KievRegime", "Zelensky Regime", "ZelenskyWarCriminal "
"Ukronato", "NATORussiaWar", "NATOWarCrimes", "Protest NATO", "NoToNATO", "Western hegemony US neocons", "BidenWar", "Proxy War Ukraine BRICS bloc", "BRICS G7", "Russia global South"

We filtered the combined dataset by English, and retain only the more prolific users (those posting more than 20 times), and more popular assertions (those referred to by more than 10 users). Then we employ two human annotators to label each tweet as pro-Russia or anti-Russia in ideology. We treat the tweets where they uniformly agree as pro-Russia to have label 1 (454 tweets), and those as anti-Russia to have label -1 (290 tweets).

After training LM-Switch on GPT-J-6B, we prompt the LM to generate on 8 topics. The selection of these 5 topics is based on ChatGPT API[9] where we ask "Please provide a list of topics that best distinguish the policitical stance of pro-Russia and anti-Russia". The list of topics is:

Table 9: Keyword for selecting Russia-related tweets.

| List of Topics |
|---|
| "Russia's annexation of Crimea" |
| "NATO expansion" |
| "Russia's regime" |
| "The recent increase in sanctions against Russia" |
| "The anti-Russian sentiment on the Internet" |

We present generations with seed 0 on these topics are listed below, constraining a minimum length of 100 and maximum length of 300.

## I.2  SCENARIO 2: MODELLING DIFFERENCES IN MEDIAS

We studies 5 controversial news events on Ground News[10], each with a pair of contrasting media article. Then we use LM-Switch learn each news article and follow the stacen to generate on the topic.

---

[8]https://huggingface.co/EleutherAI/gpt-j-6b
[9]https://chat.openai.com
[10]https://ground.news

Table 10: Generations on topic "Russia's annexation of Crimea" on pro-Russia v.s. anti-Russia.

| Stance | Generations |
|--------|-------------|
| -1.0 anti | Russia's annexation of Crimea was an invasion of Ukraine's sovereign territory, but Russia insists that Ukraine's Crimea People's Republic is legally Russian territory.
Ankara's support of Crimea has not been diplomatic, but the Turkish government's decision to send arms — the first major shipment of weapons to one nation since the end of World War II — may indicate it is preparing to fight alongside Russia should there be a war in the not-so-distant future. |
| -0.5 anti | Russia's annexation of Crimea was an act of war against Ukraine: so say thousands of Russians who were arrested after the annexation for their presence on the streets. Their fate remains unknown. As for Kiev, it has said it will be ready to send any of these prisoners to trial, even the Russian officers, provided Moscow allows it. But Putin says he isn't planning any such thing. So where is justice in this?
Russian President Vladimir Putin (R) and Crimean parliament speaker Vladimir Konstantinich after a joint session on the first day of Russia's annexation of Crimea.REUTERS/Sergei Karpukhin |
| 0 neutral | Russia's annexation of Crimea was nothing short of a geopolitical bloodbath, but in the days after the events of March 16-17, Ukrainian President Petro Poroshenko and his government took a decisive step forward.
Kicking Ukraine's Western-oriented, US-aligned government out — the Maidan revolution was partly driven by a desire to end Ukraine's alliance with Washington — was their best chance to save their beleaguered nation from catastrophe. Their first big setback was a US-sponsored coup in Kiev in late February — Washington's fingerprints are on the Kiev junta — while after Poroshenko was elected president last July, his initial policy to pursue closer economic ties with Russia, which includes cooperation on energy sector reform, had a "reset" feel. |
| 0.5 pro | Russia's annexation of Crimea was nothing short of a geopolitical earthquake: it has been the biggest geopolitical event of the year, and one that represents a fundamental challenge to the balance of power between Europe and Eurasia.
It will have deep and long-lasting consequences, not least on the economic interests of western countries, from whose business sectors Ukraine imports vast amounts of oil and gas. |
| 1.0 pro | Russia's annexation of Crimea was nothing short of a geopolitical earthquake: it has been the biggest geopolitical event of the year, and one that represents a fundamental shift in the balance of power between Europe and Eurasia. The Crimea crisis was a shock that rattled US President Barack Obama, the UN Security Council, and the NATO alliance. But it also has had a greater impact on Russia's neighbors.
Read More: `https://on.rt.com/8egy` |

**Event 1: EU to crack down on imports of Indian fuels** Times of India article: `https://timesofindia.indiatimes.com/business/india-business/top-eu-diplomat-seeks-action-against-india-on-russian-oil-ahead-of-talks/articleshow/100288784.cms`. Reuters article: `https://www.reuters.com/business/energy/borrell-urges-eu-crack-down-imports-indian-fuels-made-with-russian-o`

**Event 2: A new bill on banning "no-fault" evictions.** The Independent article: `https://www.independent.co.uk/news/uk/home-news/no-fault-eviction-ban-uk-b2340410.html`. BBC article: `https://www.bbc.com/news/uk-politics-65612842`

**Event 3: Israeli forces kill Palestinian man** VOA article: `https://www.voanews.com/a/israeli-forces-kill-palestinian-man-in-west-bank-raid-/7094609.html`. Palestine Chronicle article: `https://www.palestinechronicle.com/israeli-occupationforces-kill-palestinian-youth-near-nablus/`.

**Event 4: Stellantis' warning to change Brexit deal** The Guardian's article: `https://www.theguardian.com/business/2023/may/17/vauxhall-maker-says-brexit-deal-must-be-renegotiated-or-it-could-shut-uk-plant`. The Telegraph's article: `https://www.telegraph.co.uk/business/2023/05/17/ftse-100-markets-live-news-unemployment-rise-inflation/`.

Table 11: Generations on topic "NATO expansion" on pro-Russia v.s. anti-Russia.

| Stance | Generations |
|--------|-------------|
| -1.0 anti | NATO expansion "has nothing to do" with Europe, but Putin wants war. And while he might start war over Ukraine right away (and get "saved" by Russian soldiers), his true motives for fighting may not be limited to his 'interest' in Ukraine. And one nation could be Ukraine, while his interests are elsewhere.
In my recent interview with Alex Jones, Alex Jones talks about many things — the NWO, Obama's lack of experience — but more specifically his view of Russia and where the US. |
| -0.5 anti | NATO expansion under pressure
By Alex Lantier
4 January 2008
There is growing pressure on NATO and Washington to halt the military buildup planned for Central Asia and for Russia, which will almost certainly be followed by additional provocative US missile deployments in Europe. A Russian military expert told World Socialist Web Site correspondents that NATO states are developing a secret plan to encircle Russia with bases and troops in Central Asia.
In a Jan. 4 interview conducted by telephone, Col. Georgi A. Korniyenko, former chief of airborne forces (1995-2000), and now research director at the Strategic Analysis and Research Institute in St. Petersburg, spoke of the "political struggle for influence" over the region after the breakup of the Soviet Union, adding that there had been efforts "to increase the presence of NATO bases in Central Asia" since the 1990s. |
| 0 neutral | NATO expansion under pressure
By Alex Lantier
4 January 2008
There is growing pressure on NATO and Washington to halt the military buildup planned for Central Asia and for Russia, which will almost certainly lead to a new military confrontation.
The growing disaffection among sections of NATO from the alliance's central focus on military intervention abroad points to a significant crisis in the alliance.
The buildup in Central Asia is the biggest US military escalation since the invasion of Iraq nine years ago. It was announced in mid-December at the Chicago Summit of the NATO-led International Security Assistance Force in Afghanistan. The NATO summit also gave the green light for expansion of the alliance to a second front in the conflict with Russia, to include Georgia, Ukraine and Azerbaijan.
The US and NATO are preparing for a new confrontation with Russia in central Eurasia. The main objective of the policy of NATO expansion is to block Russia's attempts to secure its borders and its access to its energy resources. |
| 0.5 pro | NATO expansion under pressure
By Alex Lantier
4 January 2008
There is growing pressure on NATO and Washington to halt the military buildup planned for Central Asia and for Russia, which will almost certainly lead to a new military confrontation.
The growing conflict is based on preparations for NATO expansion. The Obama administration, working with its NATO allies, is moving forward with plans to expand NATO operations in Central Asia. Washington is intensifying military operations and building up the Pentagon's logistical network in the region with special attention to Azerbaijan, Kazakhstan and Kyrgyzstan. NATO and US military facilities already operate there.
This buildup is creating tensions. The expansion plan has provoked a protest storm in much of Central Asia and the rest of the non-aligned world. It is drawing international attention to the worsening global economic crisis, with calls for Washington's spending binge and budget deficits to be addressed. |
| 1.0 pro | NATO expansion under pressure
By Alex Lantier
4 January 2008
There is growing pressure on NATO and Washington to halt the military buildup planned for Central Asia and for Russia, which will almost certainly lead to a new military confrontation.
The growing conflict is based on the fact that NATO has built up its military presence in central Asia. NATO is carrying out increasingly provocative exercises in Kazakhstan and Kyrgyzstan, while announcing a new missile system to target China, including new missiles capable of reaching China's northern provinces.
It is also escalating its efforts to create a US-led alliance with Uzbekistan and Kazakhstan, and has given notice to other Central Asia nations that it is considering plans to join NATO. All these actions are increasing anxiety among Russia, which borders |

Table 12: Generations on topic "Russia's regime" on pro-Russia v.s. anti-Russia.

| Stance | Generations |
| --- | --- |
| -1.0 anti | Russia's regime under Putin has used its control over the state to crack down on dissent.
This month's elections for the Duma will be marred by Kremlin heavyweights casting their nets wide.
The main stage in any of these campaigns is a seat on the Russian parliament, the Duma, whose 650 members represent all regions – more than two-thirds of Russian territory – and whose powers are enshrined in the constitution. |
| -0.5 anti | Russia's regime under Putin has used its control over the state to crack down on dissent.
This month's elections for the Duma will be a test of Kremlin power.
The Russian president will almost certainly win a mandate to extend his stay in the Kremlin — the Russian parliament has repeatedly rubber-stamped Putin's every power grab. The opposition however seems dispirited at the thought of this "dictatorship for a dictatorship" being extended. And then there's Navalny, once again the most high-profile critic of Putin and the man who made the opposition's election campaign a reality.
Navalny was barred from the Moscow voting as a "super-voter" after the opposition organised mass protests in central Moscow following his trial earlier this year. The opposition have not been able to mount a serious campaign against Putin and his party, the United Russia — Navalny is effectively the opposition leader after a four-year hiatus. |
| 0 neutral | Russia's regime under Putin has used its control over the state to become the country's most-profitable sector. But as the economy deteriorates, that control looks increasingly fragile.
The Russian economy is facing its worst downturn in decades, with low oil prices and Western sanctions cutting off access to critical markets and funds. At the same time, Russia's central bank is trying to preserve the value of the rouble, fearing that the currency – which rose against the dollar from 46 to the dollar last year – will slump too far.
That is making for an awkward combination of economic misery and political uncertainty as the Vladimir Putin regime seeks to hold on to power. With elections coming up to parliament in a matter of months, Putin – who became president after Boris Yeltsin's 1999 resignation for health reasons – has to find ways to keep his base of support with people who lost their jobs and homes during the country's worst economic crisis since the Second World War.
Here are four steps Putin's regime is taking to do that. |
| 0.5 pro | Russia's regime under Putin has used its control over the state to become the country's most-profitable sector. But as the economy deteriorates, that control looks like it could be under threat.
It wasn't that long ago when, in the face of economic disaster, Russia found a way out of the doldrums – something that had proved much more difficult to do in the early post-Soviet years. The country's GDP plunged to $0.9 trillion last year, with more than five years of consecutive decline; the average inflation rate was 12.3 percent in September; the GDP per capita stood at $7,500. With the national currency, the ruble, collapsing, the International Monetary Fund began to express its concern and warned that a full-fledged sovereign default was a possibility.
Yet then suddenly came Vladimir Putin's economic revolution. In one brilliant stroke, the Russian leader managed to not only revive the economy but even boost its growth rate. After nearly a decade of slowdown, Putin's program has brought back Russia's GDP and, along with it, the ruble's exchange rate. |
| 1.0 pro | Russia's regime under Putin has used its control over the country's media to reinforce its image as a friendly, modern alternative to the West. Its propaganda includes news reports about its achievements in science and innovation.
It also emphasises social issues, such as promoting a healthier lifestyle, promoting marriage, and countering what it sees as Western meddling in Russia's...More..
As a result of Saturday's events, the UMP of the Russian Federation, the pro-Putin party of France, as well as France's main political parties, the PRG, the EPP |

Table 13: Generations on topic "The recent increase in sanctions against Russia" on pro-Russia v.s. anti-Russia.

| Stance | Generations |
|--------|-------------|
| -1.0 anti | The recent increase in sanctions against Russia under pressure from the European Union's (EU) and the United States have prompted Russian officials and companies involved with defense export to ask for "sanctions-defying" technologies that will allow Russia to avoid further US legislation. 
 With the Russian defense ministry estimating the cost of further foreign sanctions to be about $80 billion over 10 years and Russian companies forecasting a loss of $90 billion over the same period, a market for US-made weapons will be |
| -0.5 anti | The recent increase in sanctions against Russia under pressure from the NATO allies is another example of how the US wants to change international relations, making itself the single superpower. Its allies are already divided. As for China, it's not only the trade volume that has decreased, so have trade services," says Mikhail Prokhorov, the billionaire businessman and philanthropist. The Russian tycoon is not only a former vice-premier of Russian Prime Minister Mikhail Khodorkovsky, but also a successful businessman who is also a former oil magnate. 
 The Russian businessman is known as the "Manafort of Russia" and he is known for financing the presidential campaign to elect Donald Trump to the presidency in 2016. According to Prokhorov, this is one of the reasons for the economic war waged against Russia. "I think there could probably be a number of different reasons, and not only one. The reason is simple: we lost. We lost because we didn't understand that we were entering a period of time in which there would be a war between two groups and we, unfortunately, belonged to a third group. And today, the US controls more than 50% of the global economy – even more than China [China, in 2016 controlled 28.9% of the world's economy, according to Bloomberg] and many countries feel threatened and don't want to lose their trade volume of goods at the hands of the US," Prokhorov says. That's why they're trying to use their troops and armies to seize the Crimea from Ukraine, he says. |
| 0 neutral | The recent increase in sanctions against Russia under pressure from the NATO allies is another example of how the US wants to change international relations, making them more favorable to Washington. 
 Today's Europe is undergoing a process of reorientation; it is moving away from its historical orientation towards the Soviet Union. During his two visits to Brussels, Hungarian President Orban spoke about one of the most important aspects of his agenda: countering Russian propaganda about democracy in the EU. Brussels' and London's response indicates that they will not object too much to this Hungarian President's stance. 
 However, it is more important to note that the European policy makers are still trying to decide what role they should play in ensuring that Brussels' and London's policies succeed. It looks like a competition for power between the US and EU in terms of geopolitical influence. 
 'Russia is the problem' 
 The reason why Hungary is increasingly turning towards Russia after decades of American pressure is obvious: Budapest is well aware of its weakness. The current situation in Hungary makes it impossible for the country to compete with the US in a geopolitical confrontation. |
| 0.5 pro | The recent increase in sanctions against Russia under pressure from the NATO has resulted in an escalation of the conflict between Russia and Georgia. To counter Georgian aggression and to prepare Georgia to be able to be part of NATO when it joins the EU, Georgia has been undergoing reform of its security sector. 
 With the reform process, GAIN has seen a marked increase in the number of its staff. GAIN has expanded its activities in the reform of the Georgian army, the police and the customs service and is now engaged in the reform of the internal security services, with special attention to the National Security Service. 
 The Georgian authorities have made substantial efforts to train law enforcement personnel and to provide them with modern equipment. The reform process has gone ahead in a rather calm environment, with little public or media attention. It has been achieved with the assistance of GAIN. Georgian reforms have not been accepted by Georgia with enthusiasm but have been considered and implemented by the authorities with conviction. |
| 1.0 pro | The recent increase in sanctions against Russia under pressure from the NATO has resulted in an escalation of the conflict between the two sides. To counter the Western pressure, Russia is strengthening its nuclear "shield" by building the biggest fleet of weapons the world has seen. 
 "If the West's policies persist, one could easily envision a situation where NATO forces are stationed just on a few kilometers from the Russian borders," commented Russian President Vladimir Putin recently. |

Table 14: Generations on topic "The anti-Russian sentiment on the Internet" on pro-Russia v.s. anti-Russia.

| Stance | Generations |
|--------|-------------|
| -1.0 anti | The anti-Russian sentiment on the Internet was on full display back at the beginning of the week during a discussion on the Russian embassy in Israel on Russian television. At noon on Thursday, October 26, Channel 20's Russian-language news broadcast discussed Russian news of the week. The discussion also featured Russian Embassy social media "journalist" Sergey Pashkovsky. |
| -0.5 anti | The anti-Russian sentiment on the Internet was on display again on Monday. 
 We apologize, but this video has failed to load. tap here to see other videos from our team. Try refreshing your browser, or Google, Facebook and Twitter target pro-Kremlin users Back to video 
 Facebook, Twitter and Google were embroiled in arguments with users after they accused one another of manipulating a Russian election rally in the Ukrainian city of Kharkiv on Saturday. Each company said the other's algorithms had created a fake pro-Russian bias during the rally — which the Russian Foreign Ministry called the "biggest provocation" in years. 
 We've had to block more than 2.5 million bots |
| 0 neutral | The anti-Russian sentiment on the Internet was on display again on Monday. 
 We apologize, but this video has failed to load.tap here to see other videos from our team. Try refreshing your browser, or Google, Facebook and Twitter target pro-Kremlin users Back to video 
 Facebook, Twitter and Google were embroiled in arguments with users all over the world. Russia was a target, as well as Ukrainian-centric groups in other countries opposing the annexation of Crimea by Moscow's troops in 2014. 
 But the debate on Monday wasn't about Ukraine but about pro-Moscow supporters who use Google, Twitter and Facebook to boost the Kremlin's legitimacy and sway public opinion. 
 "We've seen cases where pro-Kremlin actors are manipulating the discourse on mainstream social media sites; in most cases, they're pushing pro-Kremlin and pro-Russian content, and some content is even pushing Russian propaganda throughout Europe," Brian Fishman, lead author of the paper that was released Monday about Russian trolls operating on Facebook, told The Canadian Press. |
| 0.5 pro | The anti-Russian sentiment on the Internet was on display again on social media last week as thousands of Russians voiced their indignation and protest over American sanctions against the former Soviet republic of Georgia during "Euromaidan," the popular nickname for demonstrations that overthrew President Mikheil Saakashvili's government in February. 
 And while much of the criticism was directed at Washington — often couched in sarcasm — other users accused the Kremlin of playing a direct hand in the overthrow as part of a long-running campaign of influence. 
 It was easy to see the echoes of last year's events that triggered the Maidan protests, which had to do with the perceived mistreatment of a pro-Moscow ethnic minority in the restive breakaway state of Ossetia that had been subjected to ethnic Georgian militia bombings and incursions from Georgia into the province. 
 The latest sanctions include restrictions against doing business in the United States with individuals and firms involved in or connected to the former Soviet republic. The US Treasury Department has slapped sanctions on 13 individuals and companies, including Giga, an international firm that provides outsourcing services. |
| 1.0 pro | The anti-Russian sentiment on the Internet was nothing short of shocking at the beginning of 2017, with popular posts ranging from "Russians are Satan" and "I will shoot them like flies" to "I'm going to kill you, Russian*" and "PunkPutin" (which also happens to be the name of a popular Russian punk rock band). This phenomenon has been blamed by some on Russian media and their allegedly fake news promoting Russophobia. However not everyone agrees with this and some argue that it is not the media propaganda but more the pro-democracy political campaigns in the US and Europe that are responsible for the Russophobia on the Web and social media. 
 Russophobia has existed for years, to be exact as far back as the 1930s when it was used by anti-communist elements in the West to isolate and demonise the Soviet Union. The term Russophobia appears to have originally been coined by the German historian Hans Kohn in the late 1930s, after it was used in the Western media |

Table 15: Generations about EU's sanction on India.

| Stance | Generations |
| --- | --- |
| Times of Indea | EU diplomat seeks action against India on at least 1 issue
The European Union's chief diplomat seeks action against India on at least 1 issue, ahead of its talks with the European Union.
"As the largest democracy and a country with the fifth largest economy, India has the means to address its growth and to address the critical issues," a diplomat said. |
| | EU diplomat warns against embracing India as a 'laundromat'
The EU diplomat said the view that Europe should embrace India to address its growth is a "laundromat" by his description and he said he will raise the issue with his counterparts in Delhi. |
| | EU diplomat Josep Borrell said the embassy will have to close with effect from Tuesday if Britain fails to take measures to avoid disruption, although the measures will not be made known before Wednesday. |
| | EU diplomat Josep Borrell has called on the UK to remain firm on its stance against India for its rising crude import from India and the UK's rising exports of refined products, as per a latest media report. |
| | EU diplomat Josep Borrell, the spokesman for the United Nations' Financial Action Task Force on money laundering. The watchdog last month called on India to act on the issue, describing India as a "laundromat" for Russian oil. |
| Reuters | EU diplomat's remarks target EU sanctions policy
The EU diplomat said that the EU should have a sanctions policy but that the sanctions rules need to be tightened to be more rigorous. |
| | EU diplomat had his visa revoked after he raised concerns he could be targeted by India's crackdown on foreigners following an attack on an Indian diplomatic post in Kabul, it emerged on Tuesday. |
| | EU diplomat said the EU would not be a willing or a competitive market to India's exporters, suggesting that it would have to use more of the talks to be a more competitive exporter to the EU member states. |
| | EU diplomat said to have met the buyer.
"The Russians have emerged as a centre where buyers of Russian oil are being routed to - including from India."
(This report was originally published on The Economic Times.) |
| | EU diplomat's remarks raise more than a few eyebrows.
"The EU diplomat said his remarks were remarks he would have made to his wife," the EU's chief diplomat said.
"But he did not refer in his remarks to Russia's invasion of Ukraine, as he said he would have said to his wife... But his remarks refer to Russia's invasion of Ukraine," the EU diplomat said. |

Table 16: Generations about banning "no-fault" eviction.

| Stance | Generations |
|---|---|
| The Independent | The new bill also seeks to impose a 20% hike in rent for several properties in London from 20th February and calls for the introduction of a private rental sector property portal to be set up. |
| | The new bill will affect how many times an assistant director will be able to refuse to rent to a black or an asian or an in this new bill or new bill will affect how many times an assistant director will be able to refuse to rent to a black or an asian or an in this new bill or new bill will affect how many times an assistant director will be able to refuse to rent to a black or an asian or an in this new bill will affect how many times an assistant |
| | The new bill will also be published on Wednesday, the government said. "The Bill seeks to bring serious offences against these crimes against women under one umbrella and make it illegal for these criminals to misuse the name of the law for their misdeeds," said government spokesperson Leela Mani. |
| | The new bill will be published in the UK parliament on 1st of April, and is said to be a "huge" opportunity to reform the rules for home landlords and for families, and is a "huge opportunity" to help families with rent". |
| | The new bill also bans so-called no-fault evictions in one of the harshest eviction bans for rental housing since the government put anti-no-fault eviction laws into law in the four years after the recession in the housing collapse of four years ago. About 20 percent of homes across the country are rented. |
| BBC | The new bill would also make it easier for landlords to repossess properties from tenants. A Conservative spokesman said the bill would make it easier for landlords to repossess properties from tenants. "We are also able to introduce a national register of landlords. We would also make tenants who use a landlord's property with their permission pay more rent. |
| | The new bill will also make it easier for landlords to refuse benefits to their tenants, which the bill would make illegal. "A huge part of our research is around what's called Section 11 - under 11 our bill makes it illegal for landlords to refuse benefits, which the bill also makes it easier for landlords to refuse benefits," Mr Donnelly said. |
| | The new bill would also be a challenge for the government, which has said it would be willing to negotiate with the union to reach an agreement. "We will have to negotiate, we will have to struggle to reach these agreements," said Conservative spokeswoman Kate Purchase said. "We will have to struggle through what is a difficult process." |
| | The new bill would also make it easier for landlords to repossess properties from anti-social tenants. A government spokesman said the bill would make sure nothing in the bill could force landlords to repossess properties from those with anti-social records. |
| | The new bill could force more landlords to be registered with the rental market and pay progressively more rental increases - which the bill intends to make easier. "We would be hoping the bill would reduce the market and reduce the supply in the rental market. It would be good for tenants to have the right to the benefits of the bill," Mr Samwell said. |

Table 17: Generations about Israel and Palestain.

| Stance | Generations |
|---|---|
| VOA | Isreal Forces and Hamas to blame for Gaza violence
"Palestinian fighters shot and killed 22 Egyptian and five Palestinian suspects from a Gaza refugee camp in a raid in Nablus early Monday. |
| | Isreal Forces and Israel-Palestinian Forces
"Israel has its forces in Israel, Palestinians have their forces in the West Bank and Gaza. Israel and Palestinians have been fighting for years and have been fighting for 26 years. Israel and Palestinians have been fighting for 26 years for security and security for security for Israelis. Palestinians were in refugee camps which were not on Palestinian areas of security. Palestinians have been in refugee camps which were not on areas of security. Palestinians were not in are as of security where |
| | Isreal Forces and the Palestinians
by Steve
(a.k.a. Eretz Israeli)
"If Palestinians want self-rule and an area for cars and families and cars and families they want to have cars and families than they have to participate in the Israeli military and they have to abide by those rules and have to have cars and families." |
| | Isreal Forces
Israel, for its part, said the forces were inside its Gaza Strip and Israeli military said suspects hurled rocks and explosives and fired at its forces in the West Bank. |
| | Isreal Forces" and its forces. They are a branch of the military of Israel. Their mission is to prepare for war.
A group of Palestinians took part in the soldiers in the area of Nablus, a group of soldiers came to the home of a Palestinian man to prepare for its defense, the Palestinian man responded to the soldiers from a member of the Palestinian forces, during the clash, the Palestinian soldiers were the people were in the home were of two brothers, in the forces |
| Palestine Chronicle | Israel Forces Kill Palestinian Youth During UN Protests
Israeli forces during a massive military raid in the West Bank on the night of July 23 to 24 during the Gaza border clashes that killed two Palestinian youth. |
| | Israel Forces Kill 26 Palestinians, Including One Child
Israeli forces raided the family house of the slain Palestinian youth east of the occupied city of Nablus, on the night of May 13-14, and killed the killed the slain youth, the family told the Washington Office on Arabic of the Israeli aggression |
| | Israel Forces Kill Palestinian Youth during the Gaza Strip Intifada
[Note: In the previous Israeli aggression against the Gaza Strip in November of 2008, the Israeli forces killed Palestinian youth during the Gaza Strip Intifada that was a natural reaction to Israeli aggression in the Gaza Strip in the last two years. |
| | Israel Forces Begin to Deploy Tanks Near the Gaza Strip
By NIDAR SINGH
JERUSALEM, (IPS) 27 Dec — Israeli forces on Monday began to concentrate tanks on the Gaza border in preparation of what Israeli government sources said was yet another aggressive invasion of the Gaza Strip. |
| | Israel Forces Kill Palestinian Youth in Bethlehem, Gaza
Palestinian youth were killed by Israeli gunfire in the occupied West Bank and Gaza Strip on the Monday to the Thursday. |

Table 18: Generations about Stellantis' warning.

| Stance | Generations |
|---|---|
| The Guardian | Vauxhall maker says Brexit deal must be renegotiated or it could shut UK plant.
The future of the British car plant which also produces Fiat Chrysler and Citroën vehicles, could depend on what deal the government's Brexit deal may be. |
| | Vauxhall maker says Brexit deal must be renegotiated or it could shut UK plant.
Many more people are employed in the UK by the UK Vauxhall and many more people will be employed in the UK by the UK Vauxhall if it is shut. |
| | Vauxhall maker says Brexit deal must be renegotiated or it could shut UK plant, putting thousands of jobs at risk.
"If we close the UK, we close the UK, that will have more impact on the UK than it will on Vauxhall and some of its employees who are not going to go just because there are others who are going to open up in the UK," the company said. |
| | Vauxhall maker says Brexit could force it to raise UK Brexit deal to unsustainable cost of electric vehicles.
If the cost of electric car batteries becomes unsustainable in the UK or the cost of batteries becomes unsustainable in Europe, it will not mean there will be no electric cars. |
| | Vauxhall maker says Brexit deal must not affect UK operations or future coronavirus production.
The future of the UK and the place of Vauxhall and many thousands of businesses after the country's decision to rejoin the trading blocks of the EU was one of the most discussed and acrimonious issues of the day. |
| The Telegraph | Vauxhall maker says workforce is currently 'at low levels' but adds that latest updates are designed to add 'at least' another 200 but that is dependent on cust omers and that they have built capacity to add. |
| | Vauxhall maker says Britain cannot rely on 'unlimited low-wage' workforce.
The British workforce is too dependent on low-wage, limited-skill, limited-skill, limited-wage migration to allow our economy to change to the skills and wage economies we need for the labour market to change for our economy to change for our people to change to change, a British manufacturer has warned. |
| | Vauxhall maker says Britain needs Brexit to end 'unlimited, low skill' migration - latest updates
The Brexit decision to change our economic model has come at the worst possible time for everyone but everyone. |
| | Vauxhall maker says it could close down"
"The Chancellor was warned by two Cabinet colleagues this morning to think twice" |
| | Vauxhall maker says this will be Britain's last model - but Britain will end one of the great two-seater car series
The decision to end the Astra has come after a decision to switch from being designed by a British company to being designed and built by a third-party, with all profits from the British model being ploughed into a new model |

**Event 5: Bulgarian Military Aid** We selcted the Russia Today news: `https://www.rt.com/news/565894-bulgaria-weapons-aid-ukraine/` and Reuters news: `https://www.reuters.com/world/europe/bulgarian-lawmakers-give-government-month-devise-ukraine-military-aid-plan-2022-11-0`

Table 19: Generations on Bulgarian Military Aid.

| Stance | Generations |
|---|---|
| Russia Today | Bulgarian President B.R. Klement has been urging President B.R. Klement to have NATO negotiate with Israel on pacifism and negotiate treaties with Ukraine until negotiations have been concluded. |
| | Bulgarian group to offer Bulgarian medical supplies to Sofia people? |
| | Bulgarian President! President Gorbachev has the most impressive equipment of the war to make good on the Soviet Union. So? The most important equipment of the war to make good on is equipment from G-2. This equipment is a huge part of Soviet equipment. |
| | Bulgarian drug users risk alcohol abusers and those who risk alcohol abusers are those exempted from mandatory intoxication tests (MADT).rehensive drinkers who supply alcohol are excluded from intoxication tests (TRIF). |
| | Bulgarian politicians are debating Bulgaria's decision to leave the European Union. Bulgaria has since a year voted to leave the European Union, which despite dragging Bulgaria into the region's conflict, has still not sold Bulgaria a NATO treaty until Bulgaria has sold Bulgaria a NATO treaty until Bulgaria has sold Bulgaria NATO treaty. |
| Reuters | Bulgarian lawmakers give lawmakers first month weeks in parliament to vote on Ukraine's decision to send only its first-class from the country. |
| | Bulgarian Soviet Socialist party
bitcoin rebate
bitcoins month trading trading swap trading weapons trading weapons Soviet Soviet Socialist party. |
| | Bulgarian economy can send one in from one month to two months quickly could be one of the main features of sending arms first. |
| | Bulgarian Soviet Union - weapons of mass-value have more military avail than Soviet Army and missile capabilities, and the Soviet military cannot not supply first-class only first-class mainly first-class Soviet military such as the SU-24 and SU-125. |
| | Bulgarian arms export sales could rise but have not spread rapidly - one way.
Heavy equipment such as what military experts have called the "forgery of the Russian army" can not be said to be for what buyers not only could probably not with it but would not be "for what one bought in Soviet Russia " |

## J   VALIDITY OF ASSUMPTIONS

To verify the validity of the assumptions, we did an experiment for searching for valid HMMs while satisfying the assumption 2. It is trivial to constrct valid $\Psi$ as long as a valid $\Phi$ can be found. So specifically, we set $d_s = 20$ and $d_c = 1$ for representing a one-conditional HMM. We let $n = 200$, and randomly initialized $\Phi$ with Gaussian distribution with variance 1e-3. Then we construct the following objective function

$$\mathcal{L} = \mathcal{L}_{\text{norm}} + \mathcal{L}_{\text{dist}} + \mathcal{L}_{\text{independence}} + \mathcal{L}_{\text{conditional}},$$

where

$$\mathcal{L}_{\text{norm}} = \sum_i \left( \sum_{s \in \mathcal{S}} \phi_{s,i}^2 - \frac{1}{dn} \sum_{s \in \mathcal{S}, i'} \phi_{s,i'}^2 \right)^2$$

$$\mathcal{L}_{\text{dist}} = \sum_{s,s'} \max(-T(s,s'), 0) + \sum_s \left( \sum_{s'} T(s,s') - 1 \right)^2$$

$$\mathcal{L}_{\text{independence}} = \sum_{i \neq j} \left( \sum_s \phi_{s,i} \phi_{s,j} \right)^2$$

$$\mathcal{L}_{\text{conditional}} = \sum_{i,j,k \text{not one value}, k \in [d_c+1, d]} \left( \sum_s \phi_{s,i} \phi_{s,j} \phi_{s,k} \right)^2$$

Generally, this objection characterizes the derivation of $\Phi$ from the assumptions. We use Adam optimizer with learning rate 1e-3, and ReduceLROnPlateau [11] with patience 100 and reduce factor 0.5. The optimization process lasts 500k steps, starting from random seeds 0, 1, 2 and 3. On all random seeds, the objective function reduces from greater than 1 to less than 1e-5. This indicates that valid HMM solutions satisfying the assumption exist.

## K    COMPARISON WITH A (SOFT) WORD BLACKLIST

First, we explain that a control vector is equivalent to a SWB. This is because by adding a vector to context $c' = c + \epsilon w$, we are equivalently adding a logit bias to each word: $c'^\top E = c^\top E + \epsilon w^\top E$, where $w^\top E$ is the static logit bias vector for each word. Then we point out that theoretically, LM-Switch is more expressive on representing sequence distributions than them, since LM-Switch's formulation $c^\top (I + \epsilon W) E$ can let different words be preferred in different contexts $c$. Theoretically, there exists a LM-Switch for switching between any two finite-length distributions (with proof below). SWB cannot achieve this (a counterexample below). Intuitively speaking, a blacklist or whitelist uniformly applied at all positions cannot possibly achieve flexible control over the complex language distribution without hurting the generation quality.

### K.1    FORMAL STATEMENT OF THE UNIVERSALITY OF LM-SWITCH

Let $D_1$ and $D_2$ be two finite-length finite-vocabulary sequence distributions, there exists a context vector function $c(o_1, o_2, \cdots, o_i)$, a word embedding $E$ and a matrix $W$, so that a LM-Switch with $-W$ and $+W$ represents distribution $D_1$ and $D_2$ respectively.

*Proof.* We prove the existence by construction. Let $I(o_1, o_2, \cdots, o_i) \in \mathbb{N}$ be an arbitrary indexing function for all subsequences. With bounded subsequence length, $I$ values are also bounded by a number $d$. We let $c(o_1, \cdots, o_i) = \sum_o \text{onehot}(I(o_1, o_2, \cdots, o_i, o)) \in \mathbb{R}^d$. For each token $o$, the word embeddings is $e_o$ such that, for any subsequence $(o_1, o_2, \cdots, o_i)$, $e_o[I(o_1, o_2, \cdots, o_i, o))] = \frac{(D_1 + D_2)(o|o_1, o_2, \cdots, o_i)}{2}$. Then $W$ is as follows: for any $(o_1, o_2, \cdots, o_i)$ and token $o$, $W[I(o_1, o_2, \cdots, o_i, o), I(o_1, o_2, \cdots, o_i, o)] = \frac{(D_2 - D_1)(o|o_1, o_2, \cdots, o_i)}{(D_1 + D_2)(o|o_1, o_2, \cdots, o_i)}$. It is diagonal. We omit verification due to space limits. $\square$

### K.2    3.2 CONSTRUCTION OF A COUNTEREXAMPLE FOR SWB

Let $D_1$ be a one-point distribution on sequence "AB", and $D_2$ be a uniform distribution on sequences "BA", "AB". For any language model representing $D_1$, there does not exist an SWB that can convert the language model into $D_2$. Verification of this counterexample is trivial, and we omit it due to space limits.

---

[11] https://pytorch.org/docs/stable/generated/torch.optim.lr_scheduler.ReduceLROnPlateau.html

## L    RESULTS OF LM-SWITCH ON PYTHIA FAMILY

Pythia [12] is a family of causal language models developed by EleutherAI. Raining in size from 14M to 2.8B, these models provide an excellent testbed for evaluating the effect of language model sizes. The table below shows the performance of LM-Switch applied to the Pythia language models. We can see a trend of better fluency but decreasing detoxification when the model size increases, indicating a higher controlling difficulty and better base quality of larger language models.

Table 20: Language model detoxification results of LM-Switch with Pythia

| Model | Toxicity↓ | | Fluency | Diversity↑ | | |
|---|---|---|---|---|---|---|
| | Max. toxicity | Toxicity prob. | Output ppl.↓ | Dist-1 | Dist-2 | Dist-3 |
| Pythia-14M | 0.208 | 0.04 | 85.67 | 0.50 | 0.84 | 0.86 |
| Pythia-70M | 0.213 | 0.06 | 54.84 | 0.55 | 0.86 | 0.87 |
| Pythia-160M | 0.223 | 0.07 | 35.24 | 0.54 | 0.86 | 0.87 |
| Pythia-410M | 0.255 | 0.13 | 36.71 | 0.58 | 0.86 | 0.86 |
| Pythia-1B | 0.286 | 0.17 | 31.34 | 0.56 | 0.85 | 0.86 |
| Pythia-1.4B | 0.289 | 0.15 | 31.63 | 0.58 | 0.86 | 0.86 |
| Pythia-2.8B | 0.328 | 0.17 | 32.90 | 0.51 | 0.81 | 0.85 |

## M    LIMITATIONS

One limitation of LM-Switch is that it works on word embeddings and focuses on conditions related to wording. This restricts its capability to deal with more complex tasks, such as syntactic trees or persuasive techniques that involve logical reasoning. Additionally, our model is dependent on word embeddings, so the model cannot work with language model APIs that do not provide direct access to these embeddings.

## N    LoRA CONFIGURATION

Thanks for suggesting a more comprehensive evaluation. We use the Huggingface Transformers' default implementation of LoRA on GPT-2-large to align with standard practices in the field and compare with our method. To test the effect of configurations, we iterate the LoRA rank over 8, 16, 32, and 64. Following the practice of Lee et al. (2023), we set the alpha scalar equal to the rank. We report LoRA's performance below.

Table 21: The performance of LoRA with different ranks on the detoxicification dataset.

| LoRA rank | Max. Toxicity | Toxicity prob. | ppl. | Dist-1 | Dist-2 | Dist-3 |
|---|---|---|---|---|---|---|
| 8 | 0.362 | 0.257 | 23.83 | 0.532 | 0.839 | 0.852 |
| 16 | 0.365 | 0.210 | 21.11 | 0.534 | 0.845 | 0.855 |
| 32 | 0.351 | 0.229 | 26.13 | 0.529 | 0.840 | 0.853 |
| 64 | 0.354 | 0.257 | 23.78 | 0.531 | 0.840 | 0.853 |

## O    INCORPORATING IN LoRA

Thanks to its theoretical foundations, LM-Switch is orthogonal to other methods and is intuitively compostable with other control methods. As an example study, we select LoRA to combine with LM-Switch on detoxification with GPT2-large as the backbone. The result is as follows. Combining LoRA with LM-Switch produces a better detoxification score than LoRA alone (although not as good as LM-Switch alone), at the cost of a degraded quality score.

---

[12]https://github.com/EleutherAI/pythia

Table 22: Results on incorporating LM-Switch with LoRA.

| Method | Max Toxicity | Toxicity prob | ppl | Dist-1 | Dist-2 | Dist-3 |
|---|---|---|---|---|---|---|
| LoRA | 0.365 | 0.210 | 21.11 | 0.53 | 0.85 | 0.86 |
| LM-Switch | 0.238 | 0.078 | 28.26 | 0.55 | 0.84 | 0.84 |
| LoRA + LM-Switch | 0.258 | 0.086 | 41.78 | 0.54 | 0.82 | 0.83 |

## P  RESULTS ON GPT-J-6B

We go on and evaluate LM-Switch on GPT-J-6B model. We use the same evaluation setup as in the main body. The results are shown in Table 23. We observe that LM-Switch is able to reduce the toxicity of the generations while maintaining the perplexity and the fluency of the generations. We also observe that the diversity of the generations is not affected by LM-Switch.

Table 23: Results of LM-Switch on GPT-J-6B model on language model detoxification.

| Method | Max Toxicity | Toxicity prob | ppl | Dist-1 | Dist-2 | Dist-3 |
|---|---|---|---|---|---|---|
| GPT-J-6B | 0.364 | 0.229 | 18.70 | 0.55 | 0.84 | 0.85 |
| GPT-J-6B + LM-Switch | 0.265 | 0.124 | 18.26 | 0.54 | 0.84 | 0.85 |

## Q  EFFECT OF LM-SWITCH ON INSTRUCTION FOLLOWING

We study the effect of LM-Switch on the LLMs' performance under prompts. For prompts, we let ChatGPT write 10 strongly positively biased prompts in Table 24. Then we let GPT-J generate 25 tokens under instructions (i.e., prompting), and compare its generation with and without LM-Switch.

Table 24: Instructions proposed by ChatGPT.

| |
|---|
| "Compose a critique of the novel 'Pride and Prejudice', emphasizing its flaws while maintaining an overall appreciative tone." |
| "Write a review of the restaurant 'Le Gourmet' focusing on areas for improvement, yet with an underlying tone of admiration for its cuisine." |
| "Pen a critique about the summer season, pointing out its drawbacks but in a way that overall celebrates its beauty and warmth." |
| "Draft a review for the TV show 'Breaking Bad', highlighting its weaker aspects but still expressing overall enthusiasm for the series." |
| "Create a review of the coffee shop 'Bean There', discussing its shortcomings while still conveying a sense of enjoyment of its atmosphere." |
| "Write an appraisal of the classic album 'Abbey Road' by The Beatles, noting any perceived faults but with a tone that remains reverent of its musical genius." |
| "Offer a critique of the play 'Hamilton', focusing on its less successful elements while still acknowledging its groundbreaking impact." |
| "Compose a review of the city of Paris in winter, pointing out the challenges of the season while still capturing the magic of the city during this time." |
| "Draft a review of the novel '1984' by George Orwell, discussing its more controversial or challenging aspects but in a context of overall admiration." |
| "Write an evaluation of the gaming console PlayStation 5, noting its limitations or flaws while still expressing enthusiasm for its technological advancements." |

We see that LM-Switch can still steer generation even under a positive prompt, while maintaining generation fluency. Without LM-Switch, the sentiment is 94.00. With a positive LM-Switch, the average sentiment increases to 97.20. With negative LM-Switch steering, the average sentiment decreases significantly to 82.80. Example generation are listed in Table 25 and 26

Table 25: Example generations (one each for the first 4 prompts) without LM-Switch

| |
|---|
| "Focus of your critique on its plot and characterization rather than its historical accuracy and literary style." |
| "(For example: not enough bread, or service was bad, or the bathrooms are dirty, or ..." |
| "In the early weeks of the season, most of the country was in the grip of an intense summer." |
| "This is a good show for all the time it spends on drug dealing but the characters are" |

Table 26: Example generations (one each for the first 4 prompts) with negative LM-Switch

| |
|---|
| "' the novel ' fails to persuade the reader that the characters'in every sense of the verb ..." |
| "The rest of the review is up to you. you should probably mention how long you've..." |
| "Tt's difficult to redeem the negative factors of summers, except for its unpredictability." |
| "Dull. It's a cheap improv comedy that's almost utterly incoherent in an..." |

## R  EMBEDDING TUNING

We also consider the possibility of tuning the word embeddings of the backbone model. We use the same training procedure as LM-Switch, but instead of tuning the last layer, we tune the word embeddings. We use the same hyperparameters as LM-Switch. The results are shown in Table 27. We observe that the performance is comparable to Soft-Blacklist, but is still inferior to LM-Switch. We believe that this is because the word embeddings are shared across all the layers, and tuning the word embeddings may cause the model to forget the knowledge learned from the pre-training. We leave the investigation of this direction to future work. Embedding-tuning is also more expensive than LM-Switch, as it requires tuning the word embeddings for the entire vocabulary, while LM-Switch only requires tuning the last layer.

Table 27: Results on embedding tuning.

| Method | Max Toxicity | Toxicity prob | ppl | Dist-1 | Dist-2 | Dist-3 |
|---|---|---|---|---|---|---|
| Embedding Tuning | 0.289 | 0.0952 | 20.41 | 0.53 | 0.84 | 0.85 |

