# OpenReview forum: "LM-Switch: Transforming Word Embedding Space for Flexible Language Model Steering"
_ICLR.cc/2024/Conference — Submitted to ICLR 2024_

### Official Review · Reviewer_dUYk · 2023-10-29

**Soundness:** 3 good
**Presentation:** 2 fair
**Contribution:** 2 fair
**Rating:** 5
**Confidence:** 3

**Summary:**

To adapt existing LLMs to diverse conditions efficiently without extensive retraining or a compromise in performance, this paper presents a lightweight method for language model conditioning named LM-Switch and provides both theoretical and empirical analysis. It applies a $d \times d $ trainable linear transformation $W$ on the output word embedding, by which the embedding $e$ of each word is replaced with $e + \epsilon W e_v$. It explains the feasibility of LM-Switch from the perspective of Hidden Markov Model, and obtains guarantees for continuous control and compositional control through its linearity properties.

**Strengths:**

1. The proposed LM-Switch is flexible and adaptable, which can be fine-tuned or adjusted to different conditions with minimal data, making it a versatile tool for various applications.
2. The author clearly demonstrated the motivation and performance of LM-Switch in LM conditioning.

**Weaknesses:**

1. There are inconsistencies in the table, some of the best metrics are highlighted in bold, while others are not (e.g. Table 1 and 2).
2. There is a formatting error in the text description of Figure 2(b).
3. The design of the baseline experiment is not well-developed and does not intuitively demonstrate the effectiveness of LM-Switch. It is necessary to provide metrics from the vanilla backbone model of LM-Switch on relevant tasks as an ablation study to validate the effectiveness of LM-Switch. A comparison of the performance between LM-Switch and directly training embedding parameters also needs to be provided.
4. Missing citations for DExperts in the main text and repeated references “Alisa Liu, Maarten Sap, Ximing Lu, Swabha Swayamdipta, Chandra Bhagavatula, Noah A. Smith, and Yejin Choi. DExperts: Decoding-time controlled text generation with experts and anti-experts”
5. It is a little hard to understand. Section 3.2 presents a bunch of fancy mathematical formulas, followed by an assumption. Then, in theorem 1, it is assumed that the assumption holds. I feel like it's not very solid. In addition, I feel like introducing the concept of Hidden Markov Models (HMM) is a bit unnecessary or overly complicated.

**Questions:**

1. In training, do you freeze the LLM parameters?
2. In section 3.3, the author said “When negative texts are available, we also fit them with M(-\epslion W)” I do not see why negative text should be fit with M(-\epslion W), since W is a learnable parameter. I hope the author can provide further explanation.

---

> ### Author Response · Authors · 2023-11-22
> **Reponse to Reviewer dUYk**
>
> We thank the reviewer for the time and review and are glad to see that the reviewer appreciates the “versatile tool” brought by LM-Switch and the clearly demonstrated motivation. Below is a detailed response to the comments.
>
> **Table Representation**
>
> Thanks for pointing out a mismarked score in Table 2. Fluency and diversity metrics are not bolded in Tables 1 and 2 following practices in prior work, as they are not the main metric in two tasks.
>
> **Figure 2a & 2b Captions**
>
> We increased the margin between the caption boxes in Figures 2a and 2b to make them clearer.
>
> **Backbone Comparisons**
>
> The reviewer raises a valuable concern for discussion. We added the backbone model performance in the revision. For the embedding training, we would like to point out that due to the large size of vocabulary, word embeddings with size $n_{vocab} \times d$ are much larger in size than LM-Switch as well as LoRA. We list its performance below. It shows comparable performance with Soft-Blacklist but is still inferior to LM-Switch. The results are added to Appendix R.
>
> | method | Max Toxicity | Toxicity prob | ppl | Dist-1 | Dist-2 | Dist-3 |
> | ----------- | ----------- | ----------- | ----------- | ----------- | ----------- | ----------- |
> | Embedding Tuning | 0.289 | 0.0952 | 20.41 | 0.53 | 0.84 | 0.85 |
>
>
> **Citation to DExperts**
>
> Thanks for pointing out the errors in citation and we corrected them in the revision.
>
> **HMM Formulation**
>
> Thanks for this valuable question. This work aims to provide theoretical motivations and understanding of LM-Switch. We select HMM as a convenient analysis tool because of its universality in modeling sequential data and wide applications in various domains. The same theorem can also migrate to multiple other frameworks, such as the Observable Operator Model (OOM) or Linear Dynamic System (LDS), with only minor modifications. We also call for new theoretical frameworks that better model the linguistic data in future works. Secondly, Assumption 1 we made in the paper is extended from properties of independent and normalized random variables. We also discuss their validity in Appendix J.
>
> **Do We Freeze Other Parameters**
>
> Yes, we freeze all other parameters in the language model, as explained in the abstract and Section 3.3.
>
> **Task Selection**
>
> In this work, we train LM-Switch to fit a steering spectrum or dimension. To this end, we should specify two points to define the steering dimension better and contrast the differences between the two directions. In detoxification, this dimension is defined by toxic v.s. non-toxic sentences, and in sentiment control,  the dimension is defined by positive and negative sentences. If we only use the train on positive texts, we observe that required to generate in the opposite direction, LM-Switch simply avoids words and topics as a shortcut solution, which yields inferior performance.
>
> Finally, we hope that this addresses your concerns. If you find our response useful in answering your questions, we kindly request you to consider updating the reviews.

---

### Official Review · Reviewer_Stxg · 2023-10-30

**Soundness:** 2 fair
**Presentation:** 2 fair
**Contribution:** 2 fair
**Rating:** 3
**Confidence:** 4

**Summary:**

The paper proposes a linear transformation of word embedding $E = E + \epsilon WE$ (called LM-Switch) that can be plugged into any LM to steer generation. LM-Switch is evaluated on 3 tasks: 	language detoxification, sentiment control, and political stance control. The evaluation shows that LM-Switch performs on par or slightly better than published results.

**Strengths:**

LM-Switch is simple to implement and has a small number of parameters..

**Weaknesses:**

The major weakness of this paper is its evaluation.

- The evaluation is too simple and unrealistic to assess the effectiveness of the proposed method LM-Switch. First, the three tasks in the evaluation are binary tasks. This allows for picking positive and negative values of e to control. Thus, it’s unclear whether the LM-Switch generalizes to non-binary tasks.

- While LM-Switch achieves a better Max Toxicity score than other models, the fact that the soft-blacklist method is doing well might suggest that the testset for toxicity is simple.	Moreover, it looks like the results from other methods are quoted in the paper instead of direct comparison by implementing those methods on the same GPT-2 base-model. This leads to unfair comparison.

- There is no human evaluation. Note that for language generation tasks, it is important to have human evaluation as we can’t trust automatic metrics. DExperts paper has human evaluation for both language detoxification and sentiment control. The political stance study in this paper is not systematic and based on some cherry pick examples. Having said that, without properly running human evaluation, it’s unclear how good LM-Switch is.

- The GPT-2 large model has only ~800M parameters, which is considered small by today's standard. Thus I do not find the argument about parameter efficiency in the paper is convincing. Why not apply the proposed method for Llama-7b or Llama-65B models?

- How does LM-Switch change the behavior on language generation after being tuned for binary tasks? The paper said LM-Switch maintains balanced generation quality but it is evaluated using only perplexity. For a language generation application, I could imagine a prompt such as “write a review criticizing a movie X but in a positive tone”, how does the model behave in such a case?


Other minor weaknesses:

- The paper claims LM-Switch is theoretically grounded by analyzing HMM. But HMM is completely different from autoregressive LM and analysis on HMM with markov assumption is not true on LM unless it’s proven directly for autoregressive LM.

**Questions:**

See questions in the weaknesses section.

---

> ### Author Response · Authors · 2023-11-22
> **Response to Reviewer Stxg (1/2)**
>
> We are pleased to see the reviewer appreciate the proposed method as “simple to implement” and parameter efficient. We thank them for the detailed reviews. Regarding the comments:
>
> **Non-binary tasks**
>
> The reviewer raises an interesting concern. First, the hyperparameter $\epsilon$ was selected based on a validation set in Appendix E. Second, about the selection of tasks, this work tackles the steering of language models. The steering problem requires a spectrum or dimension to define the problem properly. Additionally, we demonstrate the non-binary tasks via compositionality of LM-Switch in Section 4.2. Therefore, different sub-dimensions can be composed by LM-Switch to achieve complex control (e.g., to “generate non-toxic and positive comments”).
>
> **Baselines for toxicity**
>
> The soft-blacklist method can be seen as a simplified version of LM-Switch. It also serves as a surprisingly simple but useful baseline yet unexplored by prior papers on the detoxification task. The fact that its performance is reasonable while still inferior to LM-Switch implies the powerfulness of this thread of work by exploiting word embeddings. As for the backbone choices, in the revision, we added the backbone model for a fairer pairwise comparison with our model.
>
> **No human evaluation**
>
> We thank the constructive concern raised by the reviewer. We conduct a human evaluation of the detoxification task and compare LM-Switch with LoRA, DExperts, and GPT-2 in a pairwise manner. Specifically, we follow the practice in DExperts and ask 4 human annotators to compare 50 generations from LM-Switch and the baseline from 3 perspectives: detoxification, fluency, and being topical to the prompt. The results are as follows. We can see that LM-Switch is ranked significantly less toxic and more topical than the baseline. It performs similarly to DExperts and GPT-2 but better than LoRA on fluency. These results are added to Section 4.1.
>
> | Percentage | Less Toxic | More Fluent | More Topical |
> | --- | --- | --- | --- |
> | LM-Switch | **19.0** | **21.0** | **18.0** |
> | Neutral | 69.5 | 69.0 | 69.5 |
> | LoRA | 11.5 | 10.0 | 12.5 |
> | --- | --- | --- | --- |
> | LM-Switch | **24.5** | 21.0 | **32.0** |
> | Neutral | 56.5 | 57.5 | 47.0 |
> | GPT-2 | 19.0 | **21.5** | 21.0 |
> | --- | --- | --- | --- |
> | LM-Switch | **24.0** | **25.0** | **32.0** |
> | Neutral | 56.5 | 52.0 | 56.5 |
> | DExperts | 19.5 | 23.0 | 11.5 |
>
> **Parameter efficiency**
>
> As explained in Section 1, LM-Switch only uses $d \times d$ parameters. On larger models like GPT-J 6B, as they typically have 4k word embedding dimensions, the trainable parameters are only 16.7M, which is slightly smaller than a rank-32 LoRA on the same backbone model (18.4M). Its results with LM-Switch are listed below. These results are added to Appendix P.
>
> | method | Max Toxicity| Toxicity prob | ppl | Dist-1 | Dist-2 | Dist-3 |
> | ----------- | ----------- | ----------- | ----------- | ----------- | ----------- |----------- |
> | GPT-J-6B | 0.364 | 0.229 | 18.70 | 0.55 | 0.84 | 0.85 |
> | GPT-J-6B + LM-Switch | 0.265 | 0.124 | 18.26 | 0.54 | 0.84 | 0.85 |

---

> ### Author Response · Authors · 2023-11-22
> **Response to Reviewer Stxg (2/2)**
>
> **Behavior on language generation: comparison with prompting**
>
> The reviewer raises a valuable question regarding LM-Switch’s efficacy and generation quality under prompting. We let GPT-J generate 25 tokens under instructions (i.e., prompting), and compare its generation with and without LM-Switch. For prompts, we let ChatGPT write 10 similar instructions to the reviewer’s suggestion. We see that LM-Switch can still steer generation even under a positive prompt while maintaining generation fluency. These results are added to Appendix Q.
>
> |*Instructions*|
> | ----------- |
> |"Compose a critique of the novel 'Pride and Prejudice', emphasizing its flaws while maintaining an overall appreciative tone."|
> |"Write a review of the restaurant 'Le Gourmet' focusing on areas for improvement, yet with an underlying tone of admiration for its cuisine."|
> |"Pen a critique about the summer season, pointing out its drawbacks but in a way that overall celebrates its beauty and warmth."|
> |"Draft a review for the TV show 'Breaking Bad', highlighting its weaker aspects but still expressing overall enthusiasm for the series."|
> |"Create a review of the coffee shop 'Bean There', discussing its shortcomings while still conveying a sense of enjoyment of its atmosphere."|
> |"Write an appraisal of the classic album 'Abbey Road' by The Beatles, noting any perceived faults but with a tone that remains reverent of its musical genius."|
> |"Offer a critique of the play 'Hamilton', focusing on its less successful elements while still acknowledging its groundbreaking impact."|
> |"Compose a review of the city of Paris in winter, pointing out the challenges of the season while still capturing the magic of the city during this time."|
> |"Draft a review of the novel '1984' by George Orwell, discussing its more controversial or challenging aspects but in a context of overall admiration."|
> |"Write an evaluation of the gaming console PlayStation 5, noting its limitations or flaws while still expressing enthusiasm for its technological advancements."|
>
> | *Setting*  | *Sentiment Scores*|
> | ----------- | ----------- |
> |Without LM-Switch | 94.00|
> |With positive LM-Switch $\uparrow$ | 97.20 |
> |With negative LM-Switch $\downarrow$ | 82.80 |
>
> *Example generations (one each for the first 4 prompts)*
> | Without LM-Switch | With negative LM-Switch |
> | ----------- | ----------- |
> | “Focus of your critique on its plot and characterization rather than its historical accuracy and literary style.” | "the novel fails to persuade the reader that the characters'in every sense of the verb …" |
> | “(For example: not enough bread, or service was bad, or the bathrooms are dirty, or …” | “The rest of the review is up to you. you should probably mention how long you've…” |
> | “In the early weeks of the season, most of the country was in the grip of an intense summer.” | “It's difficult to redeem the negative factors of summers, except for its  unpredictability.” |
> | “This is a good show for all the time it spends on drug dealing but the characters are” | “Dull. It's a cheap improv comedy that's almost utterly incoherent in an…” |
>
> **Why HMM**
>
> Thank you for this inspiring comment on our theoretical motivation. About the selection of HMMs as a framework, this work aims to provide theoretical motivations and understanding of LM-Switch. We select HMM as a convenient analysis tool because of its universality in modeling sequential data and wide applications in various domains. The same theorem can also migrate to multiple other frameworks, such as the Observable Operator Model (OOM) or Linear Dynamic System (LDS), with only minor modifications. We also call for new theoretical frameworks that better model the linguistic data in future works. Secondly, Assumption 1 we made in the paper is extended from properties of independent and normalized random variables. We also discuss their validity in Appendix J.
>
>
> Finally, we hope that this addresses your concerns. If you find our response useful in answering your questions, we kindly request you to consider updating the reviews.

---

### Official Review · Reviewer_vGHy · 2023-10-31

**Soundness:** 3 good
**Presentation:** 3 good
**Contribution:** 3 good
**Rating:** 8
**Confidence:** 3

**Summary:**

The authors propose a novel method to control/condition language model generation by adapting the word representations for large language models (LLM). The method is based on the hidden-markov-modelling to guide word representations to a given direction (e.g. sentiment) with a linear transformation.  The main contributions are: i) method for conditioning LLM generation, ii) application of the proposed method to LM detoxification and sentiment control generation, and iii) interpretability and computational cost of the proposed method. The method shows competitive results compared to the baselines on both application tasks.

**Strengths:**

- A principled method for conditioning LLM generation.
- Clear description of background knowledge and related work needed to understand the proposed method.
- The authors perform a  comprehensive comparison of the proposed method with baselines on detoxification and sentiment control.

**Weaknesses:**

- It is not clearly defined the selection for the model's hyperparameters.
- A possible extra contribution can be the addition of a statistical significant test or uncertainty estimates of the results.

**Questions:**

Please address the following questions during the rebuttal:

- Could you elaborate on the selection and importance of hyper-parameters (e intensity)?
- Please speculate if the proposed approach can be extended (or combined) to other tuning methods for LLM, e.g. instruction tuning.

**Details Of Ethics Concerns:**

I have no concerns.

---

> ### Author Response · Authors · 2023-11-22
> **Response to Reviewer vGHy**
>
> We thank the reviewer for the inspiring comments and are pleased to see that the reviewer appreciates the methods as being “principled” and the comparison “comprehensive”.
>
> **Hyperparameter**
>
> As the method is lightweight, the only hyperparameter is the switch value. We use $\pm 5 \epsilon_0$ universally in evaluation, which is selected based on performance on the validation set on the detoxification task. For more details, we kindly refer the reviewer to Appendix E.
>
> **Statistical significance**
>
> We thank the constructive suggestions raised by the reviewer. We headed to conduct a statistical significance test on the detoxification task using 3 random seeds. Specifically, the standard deviation of the scores is shown in the following table, which we also updated in the revision:
>
> | model | Max. toxicity | Toxicity prob. |
> | ----------- | ----------- | ----------- |
> | LM-Switch + GPT2-base | 0.296 $\pm$ 0.018 | 0.129 $\pm$ 0.012 |
> | LM-Switch + GPT2-medium | 0.215 $\pm$ 0.015 | 0.059 $\pm$ 0.029 |
> | LM-Switch + GPT2-large | 0.249 $\pm$ 0.007 | 0.089 $\pm$ 0.009 |
>
>
> **Incorporating in other methods**
>
> Thanks to its theoretical foundations, LM-Switch is orthogonal to other methods and is intuitively compostable with other control methods. As an example study, we select LoRA to combine with LM-Switch on detoxification with GPT2-large as the backbone. The result is as follows, which is added to Appendix O. Combining LoRA with LM-Switch produces a better detoxification score than LoRA alone, at the cost of a degraded quality score.
>
> | method | Max Toxicity|  Toxicity prob | ppl. | Dist-1 | Dist-2 | Dist-3 |
> | ----------- | ----------- | ----------- | ----------- | ----------- | ----------- | ----------- |
> | LoRA | 0.365 | 0.210 | 21.11 | 0.53 | 0.85 | 0.86 |
> | LM-Switch | 0.238 | 0.078 | 28.26 | 0.55 | 0.84 | 0.84 |
> | LoRA + LM-Switch | 0.258 | 0.086 | 41.78 | 0.54 | 0.82 | 0.83 |

---

> > ### Comment · Reviewer_vGHy · 2023-11-23
> >
> > Thank you for addressing my questions. I have no further comments.

---

### Official Review · Reviewer_sbje · 2023-11-06

**Soundness:** 2 fair
**Presentation:** 3 good
**Contribution:** 3 good
**Rating:** 5
**Confidence:** 4

**Summary:**

This paper proposes an approach for controlled text generation called “LM-switch”. This approach modifies the output embedding matrix (the one that produces logits given context) by adding a linear perturbation which is parametrized by a matrix W, that is learned by finetuning the perturbed language model on text satisfying the desired control variable. This approach is empirically compared against other controlled generation approaches that also involve finetuning on domain data like DAPT, DExperts, LoRA etc. on controlled generation tasks like sentiment-controlled generation and toxicity reduction. This approach is also applied to generation of text controlled by political stance.

**Strengths:**

– The paper is well-organized and easy to understand.

– The proposed technique is simple to implement and shows promising results.

– The proposed approach achieves the target attribute better than the baselines considered.

– The interpretability and transfer analysis is interesting and hints at the effectiveness of the proposed approach.

**Weaknesses:**

– The baselines appear to be disadvantaged/weak. For example, the approach reports results on GPT-2 base, medium, and large sizes but the baselines, many of which are GPT2-based seem to not be implemented under different GPT-2 sizes.

– Related to above, details of LoRA are not provided. There are many implementation possibilities and options for LoRA based finetuning but I am not sure from the writeup if this aspect was tuned to get a strong LoRA baseline.

– The paper only performs quantitative comparison on two surface-level controlled generation tasks. Although this is mentioned in the appendix, it does not consider other controlled generation tasks, especially the ones which require manipulation of deeper attributes in language.

– MuCoLa is tested as a baseline for detoxification but not for sentiment-controlled generation.

– Although the paper emphasizes that the proposed approach makes it easy to compose different control attributes, I am unable to find adequate evidence of such compositional control abilities in the results. Relatedly, figure 2b is difficult to understand and I am not sure how exactly it relates to compositional control abilities.

– The connection of neural autoregressive LMs to HMMs is tenuous but the manuscript overstates this relationship. Practically, finite state HMMs are not as expressive as neural LMs. Moreover, finding a clean transformation of the HMM state space to a neural autoregressive LM’s vectors is non-trivial and typically intractable. Therefore, the motivation of the approach via HMM hidden state representation feels forced and disconnected. Moreover, the assumptions underlying the theorems are too unrealistic. Unless I missed something, more convincing evidence should be provided to justify the validity of the assumptions.

– Assumption 1, eqn 2: what does the variable “h” mean?

**Questions:**

Please address the concerns in the review above.

---

> ### Author Response · Authors · 2023-11-22
> **Response to Reviewer sbje**
>
> We thank the reviewer for the detailed and constructive comments. We are glad that the reviewer finds the paper “well-organized” and the results “promising”. Regarding your comments:
>
> **GPT-2 Sizes**
>
> The selection of using different sizes of GPT2 is for a fairer comparison with baselines of different backbones. Following DExperts’ practice, we evaluate using backbones of GPT2-base to GPT-large so that pairwise comparison with baselines is possible. In the revision, we added the backbone size of baselines in Table 1, so that such a comparison is more straightforward.
>
> **LoRA details**
>
> Thanks for suggesting a more comprehensive evaluation. We use the Huggingface Transformers’ default implementation of LoRA on GPT-2-large to align with standard practices in the field and compare with our method. To test the effect of configurations, we iterate the LoRA rank over 8, 16, 32, and 64. Following Lee et al. (https://arxiv.org/pdf/2308.07317), we set the alpha scalar equal to the rank. We report LoRA’s performance below. We see that LoRA’s performance remains stable to the rank and still uniformly performs worse than our model. We add this discussion to the revision Appendix N.
>
> | LoRA rank | Max Toxicity | Toxicity prob | ppl | Dist-1 | Dist-2 | Dist-3 |
> | ----------- | ----------- | ----------- | ----------- | ----------- | ----------- |  ----------- |
> | 8 | 0.362 | 0.257 | 23.83 | 0.532 | 0.839 | 0.852 |
> | 16 | 0.365 | 0.210 | 21.11 | 0.534 | 0.845 | 0.855 |
> | 32 | 0.351 | 0.229 | 26.13 | 0.529 | 0.840 | 0.853 |
> | 64 |0.354 | 0.257 | 23.78 | 0.531 | 0.840 | 0.853 |
>
>
> **Evaluation Tasks**
>
> In this work, we focus on the steering of language models. This requires a continuous spectrum or dimension for defining the task and achieving fine-grained steering like continuous and compositional control. To fit the task of steering better, LM-Switch shows the capability of compositional control in Section 4.2 so that sub-dimensions of complex control tasks can be composed to achieve multi-dimension steering (e.g., to “generate non-toxic and positive comments”).
>
> **MuCoLa**
>
> Thanks for pointing out this concern. The original MuCoLa paper was not evaluated in the same setting as ours, and the MoCuLa model has a prolonged decoding process (taking ~ 3000 hours or 5 months on an A100 GPU to finish the sentiment task). Therefore, this model is hard to compare fairly with other models with academic resources. Their own paper only evaluated a subset of the sentiment dataset.
>
> **Compositionality**
>
> For compositionality, we add the control matrices from both the toxicity and sentiment tasks. In Figure 2b, when varying the sentiment control scalar (along the long axis), we see a color change indicating prone-negative (blue) switching to prone-positive (red). Independently, when toxicity control increases (the short axis), the height indicates the toxicity level from high to low.
>
> **HMM**
>
> The reviewer raised a valuable question for discussion. About the selection of HMMs as a framework, this work aims to provide theoretical motivations and understanding of LM-Switch. We select HMM as a convenient analysis tool because of its universality in modeling sequential data and wide applications in various domains. The same theorem can also migrate to multiple other frameworks, such as the Observable Operator Model (OOM) or Linear Dynamic System (LDS), with only minor modifications. We also call for new theoretical frameworks that better model the linguistic data in future works. Secondly, Assumption 1 we made in the paper is extended from properties of independent and normalized random variables. We also discuss their validity in Appendix J. We agree that the length of such discussions can be reduced to better balance theoretical and experimental components. We made this change in the revision.
>
> **Meaning of $h$**
>
> Thanks for pointing out a typo where $h$ should written as $s$. We corrected this in the revision.
>
>
> We hope this addresses your concerns and that you could kindly consider updating the review scores.

---

### Author Response · Authors · 2023-11-23
**General Responses to Reviewers**

We thank all reviewers for their detailed and valuable reviews.

We are encouraged that the reviewers appreciate our work as it is
(1) simple and light-weighted (sbje, Stxg, dUYk)
(2) but effective with promising results (sbje, vGHy, dUYk)
(3) and clear representation (sbje, vGHy).

Our key insight in this work is to introduce a “switching value” to steer the generation of language models via a lightweight, efficient and versatile conditioning. It successfully supports multi-dimensional control via compositionality. The model is both theoretically grounded (based on modeling the effect of condition shifts in HMM) and empirically effective (based on experiments on language detoxification, sentiment control, and political stance control).

In the revision, we largely updated the paper to address the concerns of each reviewer with substantial discussions, additional experiments and ablation studies added in the main paper and the Appendix, including more baselines, comparison with various prompting, incorporating in other methods, human evaluation, etc. We hope our responses addressed the concerns of reviewers. We are looking forward to discussing with reviewers on any further questions.

---

### Meta-Review · Area_Chair_kg2T · 2023-12-06

**Metareview:**

The paper proposes LLM-switch that steers an LLM by transforming the output embeddings, which is inspired by analyzing an HMM. Overall, results show that the model achieves decent performance in certain controlled text generation tasks (such as detox, sentiment control) by tuning a very small portion of parameters.

Overall, reviewers found the paper interesting and the proposed method has certain justifications. However, reviewers also pointed out that the connection between HMM and LLM is weak, and that the experimental setup is weak.

**Justification For Why Not Higher Score:**

The theoretical justification (connection between HMM and LLM) is weak and experimental setups are weak too

**Justification For Why Not Lower Score:**

N/A

---

### Decision · Program_Chairs · 2024-01-16

Reject